# Relative enrichment of ammonium and its impacts on open-ocean phytoplankton community composition under a high-emissions scenario

Pearse J. Buchanan[1,2,3], Juan J. Pierella Karlusich[4,5], Robyn E. Tuerena[6], Roxana Shafiee[7], E. Malcolm S. Woodward[8], Chris Bowler[9,10], and Alessandro Tagliabue[2]

[1]CSIRO Environment, Hobart, 7004, Australia.
[2]Department of Earth, Ocean and Ecological Sciences, University of Liverpool; Liverpool, L69 3GP, UK.
[3]Department of Global Ecology, Carnegie Institution for Science; Stanford, CA, 94305, USA.
[4]FAS Division of Science, Harvard University, Cambridge, MA, 02138, USA.
[5]Department of Biology, Massachusetts Institute of Technology, Cambridge, MA, 02139, USA.
[6]Scottish Association for Marine Science; Dunstaffnage, Oban, PA37 1QA, UK.
[7]Center for the Environment, Harvard University, Cambridge, MA, 02138.
[8]Plymouth Marine Laboratory; Plymouth, PL1 3DH, UK.
[9]Institut de Biologie de l'École Normale Supérieure, Département de Biologie, École Normale Supérieure, CNRS, INSERM, Université de Recherche Paris Sciences et Lettres, Paris, France.
[10]CNRS Research Federation for the study of Global Ocean Systems Ecology and Evolution, FR2022/Tara Oceans GEOSEE, Paris, France.

*Correspondence to*: Pearse J. Buchanan (pearse.buchanan@csiro.au)

**Abstract.** Ammonium ($NH_4^+$) is an important component of the ocean's dissolved inorganic nitrogen (DIN) pool, especially in stratified marine environments where intense recycling of organic matter elevates its supply over other forms. Using a global ocean biogeochemical model with good fidelity to the sparse $NH_4^+$ data that is available, we project increases in the $NH_4^+$:DIN ratio in over 98% of the ocean by the end of the 21st century under a high-emission scenario. This relative enrichment of $NH_4^+$ is driven largely by circulation changes, and secondarily by warming-induced increases in microbial metabolism, as well as reduced nitrification rates due to pH decreases. Supplementing our model projections with geochemical measurements and phytoplankton abundance data from *Tara* Oceans, we demonstrate that shifts in the form of DIN to $NH_4^+$ may impact phytoplankton communities by disadvantaging nitrate-dependent taxa like diatoms while promoting taxa better adapted to $NH_4^+$. This could have cascading effects on marine food webs, carbon cycling, and fisheries productivity. Overall, the form of bioavailable nitrogen emerges as a potentially underappreciated driver of ecosystem structure and function in the changing ocean.

## 1 Introduction

The chemical species of dissolved inorganic nitrogen (DIN) are fundamental for the growth of marine primary producers that underpin oceanic food webs, fisheries production and the carbon cycle. Bioavailable DIN is composed of different forms, principally nitrate ($NO_3^-$), nitrite ($NO_2^-$) and ammonium ($NH_4^+$). Typically, $NO_3^-$ is regarded as the main form. This is not

without reason, since $NO_3^-$ represents most of the total DIN stock and is prevalent in highly productive regions where it tends to fuel the majority of primary production (Dugdale, 1967). However, $NH_4^+$ and $NO_2^-$ are recognized as critical fuels for marine primary production in stratified environments, where intense recycling of organic matter can elevate their use by phytoplankton (Clark et al., 2008; Dugdale and Goering, 1967; Fawcett et al., 2011; Rodgers et al., 2024; Yool et al., 2007) and fuel rapid rates of primary production via rapid recycling even if the standing stock of DIN is low (Bender and Jönsson, 2016; Matsumoto et al., 2016; Rii et al., 2016; Yang et al., 2019).

The relative speciation of DIN plays a crucial role in shaping marine phytoplankton community composition. Marine diatoms, for instance, show a competitive edge over other types of phytoplankton for growth on $NO_3^-$ as a source of bioavailable nitrogen (Berg et al., 2003; Fawcett et al., 2011; Glibert et al., 2016a; Klawonn et al., 2019; Litchman, 2007; Van Oostende et al., 2017; Selph et al., 2021; Tungaraza et al., 2003; Wan et al., 2018). One theory posits that their ecological success in turbulent, high $NO_3^-$ environments (Margalef, 1978) may be due to a capacity to store $NO_3^-$ in their vacuoles and then rapidly reduce it when they experience sudden increases in light, which would position diatoms to rapidly consume any excess reductant that would otherwise retard growth (Glibert et al., 2016a; Lomas and Glibert, 1999; Parker and Armbrust, 2005). Meanwhile, other phytoplankton types such as cyanobacteria, more adapted to stable conditions, are considered better competitors for the reduced forms of nitrogen, including $NH_4^+$ (Fawcett et al., 2011; Glibert et al., 2016a; Litchman et al., 2007) (Fig. 1). There is intense competition for $NH_4^+$ since nitrogen in this form can be most efficiently converted into glutamate and other basic building blocks for biomass synthesis, while $NO_2^-$ and $NO_3^-$ must be reduced first within the cell (Dortch, 1990). Thus, phytoplankton types with superior affinities for $NH_4^+$, like cyanobacteria, tend to displace other taxa under nitrogen limiting conditions (Litchman et al., 2007). These competitive outcomes are also well documented in freshwater and brackish systems (Andersen et al., 2020; Carter et al., 2005; Donald et al., 2013; Örnólfsdóttir et al., 2004; Trommer et al., 2020) and appear somewhat universal in aquatic environments.

As anthropogenic pressures increase, several factors may tip the balance towards $NH_4^+$ and other reduced forms of DIN (Fig. 1). Physical changes, including a changing oceanic circulation (Sallée et al., 2021), are expected to limit inputs of $NO_3^-$ from deeper waters to further intensify nitrogen limitation of phytoplankton communities (Bopp et al., 2005; Buchanan et al., 2021). Climate warming is expected to accelerate the metabolism of phytoplankton (Anderson et al., 2021; Eppley, 1972) and thereby increase nitrogen demand and recycling rates (Cherabier and Ferrière, 2022) to potentially elevate reduced forms of nutrients in the lower latitudes (Rodgers et al., 2024). Meanwhile, ocean acidification may decelerate rates of microbial ammonia oxidation, the first step of nitrification (Beman et al., 2011). While it is unlikely that ammonia oxidation would be slowed to the point where substantial quantities of $NH_4^+$ do not undergo oxidation, a slight deceleration in the upper ocean may elevate the supply ratio of $NH_4^+$ to $NO_3^-$. These changes are expected to increase the relative availability and/or supply of $NH_4^+$ compared to the more oxidized forms of $NO_2^-$ and $NO_3^-$. Due to the intense competition for $NH_4^+$ and resulting shifts towards smaller, more competitive phytoplankton taxa, the relative enrichment in $NH_4^+$ may become a self-sustaining regime unless

new inputs of $NO_3^-$ are sufficient to reverse it. However, the magnitude of $NH_4^+$ enrichment and its dominant drivers remain unquantified. Moreover, even though there are numerous localized studies that showcase how phytoplankton taxa shift in response to changes in the composition of DIN (Berg et al., 2003; Fawcett et al., 2011; Glibert et al., 2016a; Klawonn et al., 2019; Litchman, 2007; Van Oostende et al., 2017; Selph et al., 2021; Tungaraza et al., 2003; Wan et al., 2018), we lack a general understanding of the degree to which phytoplankton communities are affected by the relative enrichment of $NH_4^+$ at the global scale. This represents an important knowledge gap as to how climate change will affect the upper ocean nitrogen cycle and phytoplankton community composition, with possible implications for carbon export and fisheries productivity.

In this work, we use a global ocean-biogeochemical model equipped with an advanced nitrogen cycle to quantify the relative enrichment of $NH_4^+$ within DIN in a future ocean. Hereafter, we use the $NH_4^+$ to dissolved inorganic nitrogen ratio ($NH_4^+$:DIN), where DIN = $NH_4^+ + NO_2^- + NO_3^-$, as a measure of this relative availability in the form of nitrogen. When we refer to the relative enrichment of $NH_4^+$, we specifically mean an increase in the amount of DIN that is $NH_4^+$, with an enrichment consistent with a higher proportion of primary production supported through regeneration (i.e., $NH_4^+$-fuelled). We comment on the potential ecological importance of this enrichment by using compilations of phytoplankton relative abundance data collected during the *Tara* Oceans expeditions and idealized experiments that isolate the effect of competition for $NH_4^+$ from $NO_3^-$.

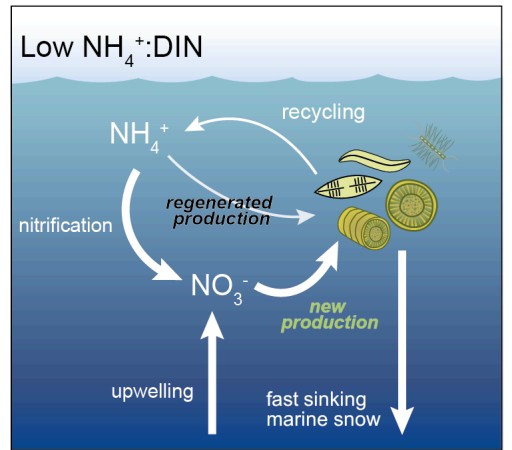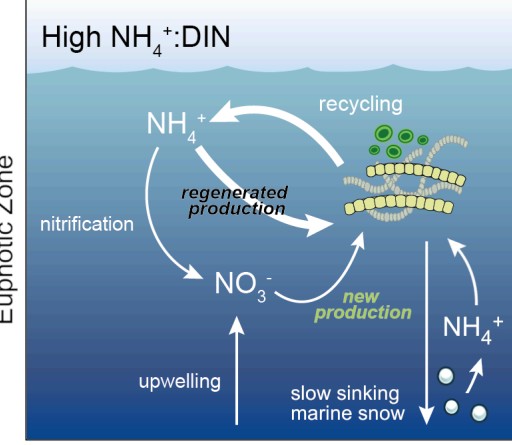

**Figure 1. Regimes of low $NH_4^+$:DIN and higher $NH_4^+$:DIN regimes in the upper ocean.** In a low $NH_4^+$:DIN regime, there is more vertical delivery of $NO_3^-$ to the upper ocean through physical mixing, which is taken up by larger phytoplankton (termed new production), including diatoms, to produce larger, denser aggregates of sinking organic matter (marine snow) that undergo recycling deeper in the water column. In the high $NH_4^+$:DIN regime, less $NO_3^-$ is mixed into the upper ocean and $NH_4^+$ supports a greater proportion of primary production (termed regenerated production). Those phytoplankton that are competitive for $NH_4^+$ tend to be smaller, and form less dense aggregates that sink more slowly. Consequently, more organic matter is recycled within the photosynthetically active zone (herein defined as where phytoplankton biomass > 0.1 mmol C m$^{-3}$) and there is more regenerated production. All processes are affected by changes to seawater properties driven by large-scale climate change.

## 2 Materials and Methods

### 2.1 The biogeochemical model

The biogeochemical model is the Pelagic Interactions Scheme for Carbon and Ecosystem Studies version 2 (PISCES-v2), which is detailed and assessed in Aumont et al. (2015). This model is embedded within version 4.0 of the Nucleus for European Modelling of the Ocean (NEMO-v4.0). We chose a 2° nominal horizontal resolution with 31 vertical levels with thicknesses ranging from 10 meters in the upper 100 meters to 500 meters below 2000 meters. Due to the curvilinear grid, horizontal resolution increases to 0.5° at the equator and to near 1° poleward of 50°N and 50°S.


We updated the standard PISCES-v2 (Aumont et al., 2015) for the purposes of this study, specifically by adding $NO_2^-$ as a new tracer. The PISCESv2 biogeochemical model already resolved the pools of $NH_4^+$, $NO_3^-$, dissolved oxygen, the carbon system, dissolved iron, phosphate, two kinds of phytoplankton biomass (nanophytoplankton and diatoms), two kinds of zooplankton biomass (micro- and meso-zooplankton), small and large pools of particulate organic matter, and dissolved organic matter
(Aumont et al., 2015). While the model does not strictly represent picophytoplankton, implicit variations in the average cell size of the nanophytoplankton type affect nutrient uptake dynamics and may therefore encompass some functionality of picophytoplankton in oligotrophic systems (Aumont et al., 2015). The addition of $NO_2^-$ necessitated breaking full nitrification ($NH_4^+ \rightarrow NO_3^-$) into its two steps of ammonia ($NH_4^+ \rightarrow NO_2^-$) and nitrite oxidation ($NO_2^- \rightarrow NO_3^-$). Both steps were simulated implicitly by multiplying a maximum growth rate by the concentration of substrate and limitation terms representing the effect
of environmental conditions to return the realized rate. For ammonia oxidation, limitations due to substrate availability, light and pH determined the realised rate. For nitrite oxidation, limitations due to substrate availability and light affected the realised rate. All parameter choices were informed by field and laboratory studies, and a detailed description is provided in the Supplementary Text S1.

New nitrogen is added to the ocean via biological nitrogen fixation, riverine fluxes, and atmospheric deposition. Nitrogen fixation and static riverine additions are equivalent to that presented in Aumont et al. (2015) and atmospheric deposition is maintained at preindustrial rates according to Hauglustaine et al. (2014) and applied as in Buchanan et al. (2021). Nitrogen is removed from the ocean via denitrification, anaerobic ammonium oxidation (anammox) and burial. The internal cycling of nitrogen involves assimilation by phytoplankton in particulate organic matter, grazing and excretion by zooplankton,
solubilization of particulates to dissolved organics, ammonification of dissolved organic matter to $NH_4^+$, followed by nitrification of $NH_4^+$ and $NO_2^-$ via ammonia oxidation and nitrite oxidation (Supplementary Text S1).

## 2.2 Model experiments

### 2.2.1 Identifying anthropogenic drivers

To quantify the impact of anthropogenic activities on $NH_4^+$:DIN ratios, we performed transient simulations by forcing the biogeochemical model with monthly physical outputs (temperature, salinity, ocean transports, short wave radiation and wind speeds) produced by the Institut Pierre-Simon Laplace Climate Model 5A (Dufresne et al., 2013). Simulations included a preindustrial control (years 1850 to 2100) where land-use, greenhouse gases and ozone remained at preindustrial conditions, and a climate change run (years 1850 to 2100) where these factors changed according to historical observations from 1850 to

2005 and according to the high emissions Representative Concentration Pathway 8.5 from 2006 to 2100 (RCP8.5) (Riahi et al., 2011). We chose a high emissions scenario to emphasize the clearest degree of anthropogenic changes and thus maximize anthropogenic effects. However, we acknowledge that the RCP8.5 is considered an extreme scenario under present development pathways (Riahi et al., 2017).

In addition, we performed parallel experiments (years 1850 to 2100) that isolated the individual effects of our three anthropogenic stressors: a changing circulation ("Phys"), warming on biological metabolism ("Warm"), and acidification effects on ammonia oxidation ("OA"). The experiment with all anthropogenic effects was termed "All". These experiments involved altering the factor of interest in line with the historical and RCP8.5 scenario while holding the other factors at their preindustrial state. Experiment "Phys", for example, involved changing the ocean's circulation, temperature and salinity, and

the resulting effects to light associated with sea-ice extent changes, but the ecosystem component of the model experienced only the preindustrial temperature, and atmospheric $CO_2$ was held at a preindustrial concentration of 284 ppm. In contrast, experiment "Warm" maintained the preindustrial climatological ocean state and atmospheric $CO_2$ at 284 ppm, but ensured that the ecosystem component saw increasing temperatures (T in ºC) according to the RCP8.5 scenario, which scaled growth of phytoplankton types according to $1.066^T$ and heterotrophic activity (grazing and remineralisation) according to $1.079^T$

(Aumont et al., 2015). Experiment "OA" held the circulation and temperature effects on metabolism constant but involved the historical and future projected increase in atmospheric $CO_2$. This decreased pH and negatively affected rates of ammonia oxidation at a rate consistent with field measurements (Beman et al., 2011; Huesemann et al., 2002; Kitidis et al., 2011), specifically a loss of ~20% per 0.1 unit decrease in pH below 8.0 (Fig. S1).

The effect of climate change at the end of the 21st century (mean conditions 2081-2100) was quantified by comparing with the preindustrial control simulation (also mean conditions 2081-2100). This preindustrial control simulation was run parallel to the climate change simulations (i.e., 1850-2100), but without anthropogenic forcings. This allowed a direct comparison to be made between experiments at the end of the 21st century and eliminated the effect of model drift. We calculated changes at each grid cell by averaging over the upper ocean where primary production was active, which we hereafter refer to as the

photosynthetically active zone defined as those depths where total phytoplankton biomass was greater than 0.1 mmol C m$^{-3}$.

In addition, we compared the preindustrial simulation with observations to explore broad patterns in $NH_4^+$ and $NH_4^+$:DIN ratios.

### 2.2.2 Isolating the effect of competition for $NH_4^+$

A unique aspect of the PISCESv2 biogeochemical model is that it weights uptake of $NH_4^+$ over $NO_3^-$ when both substrates are

low, but as $NO_3^-$ becomes abundant, the community switches towards using $NO_3^-$ as a primary fuel (Fig. 2). This is achieved via

$$l_{PFT}^{NH_4^+} = \frac{[NH_4^+]}{[NH_4^+] + K_{PFT}^N} \tag{1}$$

$$l_{PFT}^{NO_x^-} = \frac{[NO_2^-] + [NO_3^-]}{[NO_2^-] + [NO_3^-] + K_{PFT}^N} \tag{2}$$

$$l_{PFT}^{DIN} = \frac{[NH_4^+] + [NO_2^-] + [NO_3^-]}{[NH_4^+] + [NO_2^-] + [NO_3^-] + K_{PFT}^N} \tag{3}$$

$$L_{PFT}^{NH_4^+} = \frac{5 \cdot l_{PFT}^{DIN} \cdot l_{PFT}^{NH_4^+}}{l_{PFT}^{NO_3^-} + 5 \cdot l_{PFT}^{NH_4^+}} \tag{4}$$

$$L_{PFT}^{NO_x^-} = \frac{l_{PFT}^{DIN} \cdot l_{PFT}^{NO_x^-}}{l_{PFT}^{NO_3^-} + 5 \cdot l_{PFT}^{NH_4^+}} \tag{5}$$

Where $K_{PFT}^N$ is the prescribed half-saturation coefficient for uptake of inorganic nitrogen for a given phytoplankton functional type (PFT); $[NH_4^+]$, $[NO_2^-]$, and $[NO_3^-]$ are the molar concentrations of ammonium, nitrite and nitrate; $l_{PFT}^{NH_4^+}$, $l_{PFT}^{NO_x^-}$ and $l_{PFT}^{DIN}$ are the michaelis-menten uptake terms for $NH_4^+$, inorganic oxidised nitrogen (the sum of $NO_2^-$ and $NO_3^-$), and DIN; and $L_{PFT}^{NH_4^+}$ and

$L_{PFT}^{NO_x^-}$ are the growth limitation factors on $NH_4^+$ and inorganic oxidised nitrogen. In the above, the resulting $L_{PFT}^{NH_4^+}$ and $L_{PFT}^{NO_x^-}$ terms (Eqs. 4-5) are influenced by a factor 5 that is applied to $l_{PFT}^{NH_4^+}$. This assumes that $NH_4^+$ uptake is weighted five times more than oxidised inorganic nitrogen, which represents the well-established preference for growth on $NH_4^+$ (Dortch, 1990). However, as oxidised nitrogen (hereafter $NO_3^-$) becomes more abundant than $NH_4^+$, the $L_{PFT}^{NO_x^-}$ term exceeds $L_{PFT}^{NH_4^+}$, meaning that phytoplankton switch to new production over regenerated production (see cross over points between solid and dashed lines in

Fig. 2).

These dynamics are common to both PFTs: nanophytoplankton and diatoms (Fig. 2). However, a key difference is that the $K_{PFT}^N$ of diatoms is prescribed as 3-fold greater than that of nanophytoplankton, reflecting their greater average size. As a result, diatoms are always less competitive than nanophytoplankton for $NH_4^+$ and are less competitive for $NO_3^-$ when $NO_3^-$ is

scarce. However, a low $l_{PFT}^{NH_4^+}$ for diatoms also results in a higher $L_{PFT}^{NO_x^-}$ as $NO_3^-$ concentrations rise. This is evident in Figure 2,

where growth by diatoms on $NO_3^-$ (black solid line) overtakes growth by nanophytoplankton on $NO_3^-$ (green solid line) as $NO_3^-$ becomes abundant. As a result, the model gives diatoms a competitive advantage over nanophytoplankton that accords with theorized growth advantages under high $NO_3^-$ (Glibert et al., 2016a; Lomas and Glibert, 1999; Parker and Armbrust, 2005). Additionally, the switch from regenerated to new primary production occurs at much lower concentrations of $NO_3^-$ for diatoms, aligning with fields studies that identify diatoms as responsible for the majority of $NO_3^-$ uptake in the nitracline (Fawcett et al., 2011).

We sought to isolate the impact of competition for $NH_4^+$ and thus target the causative relationship between $NH_4^+$:DIN and variations in PFT relative abundance. To do so, we repeated the set of experiments described above (All, Phys, Warm, OA and the preindustrial control) from years 1850 to 2100 but with an alternative parameterization where diatoms were made to have the same growth limitation on $NH_4^+$ as other phytoplankton, so that there was zero competitive advantage or disadvantage for $NH_4^+$ between these groups (i.e., making the dashed black and green lines in Figure 2 the same under all conditions). These simulations were called "model$_{compete}$" and were initialised from the same conditions as those done with the default parameterisation, which we call "model$_{control}$". All other traits remained unchanged, including the competitive advantage of diatoms at high $NO_3^-$ but also their competitive disadvantage at low $NO_3^-$ (Fig. 2). In other words, when DIN was low, diatoms were equally competitive for $NH_4^+$ as nanophytoplankton, but still suffered their unique limitations associated with $NO_3^-$, light, silicate, phosphate, and iron availability, as well as grazing pressure, and this isolated the direct effect of competition for $NH_4^+$.

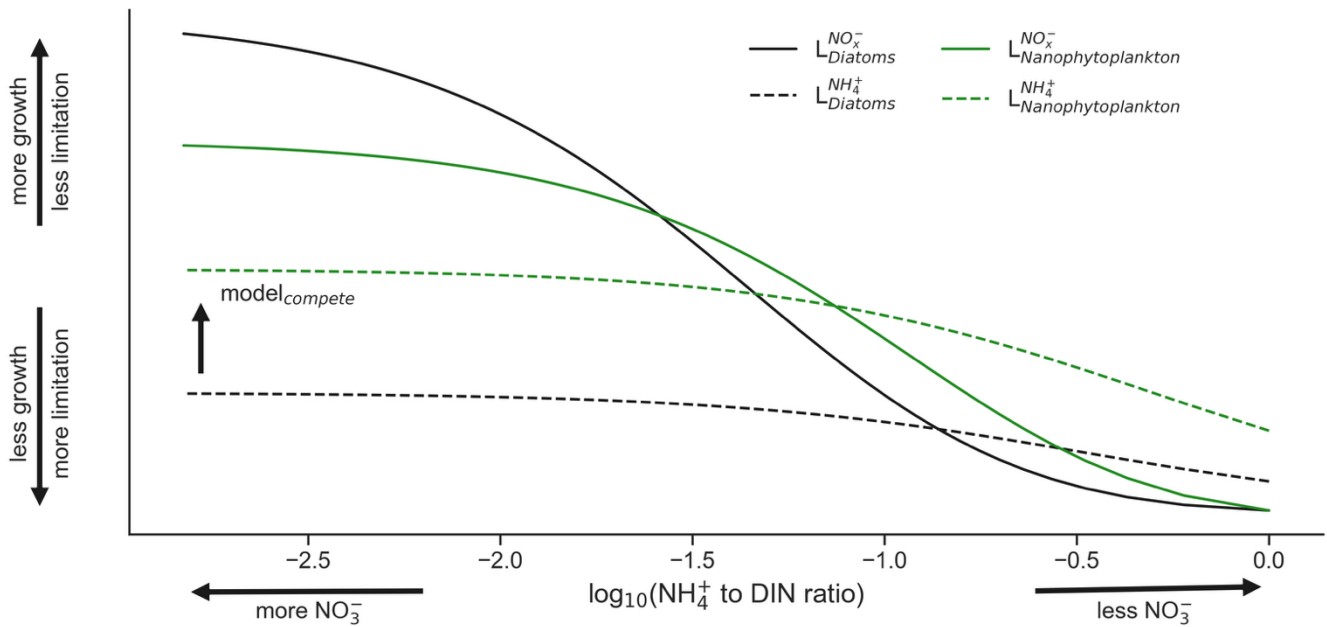

**Figure 2. Limitation of the diatom (black) and nanophytoplankton (green) phytoplankton functional types (PFT) in the ocean-biogeochemical model by $NO_3^-$ (solid lines) and $NH_4^+$ (dashed lines) as a function of the $NH_4^+$:DIN ratio on a $log_{10}$ scale.** Note that the nanophytoplankton PFT is always more competitive for $NH_4^+$ and is more competitive for $NO_3^-$ when $NO_3^-$ is low, while diatoms become more competitive for $NO_3^-$ when $NO_3^-$ is high.


## 2.3 Nutrient and rate data

Measured $NH_4^+$ concentrations (N=692; μM) were used for model-data assessment (Fig. 3; Fig. S2-S3). Nutrients were collated from published work (Buchwald et al., 2015; Mdutyana et al., 2020; Newell et al., 2013; Raes et al., 2020; Santoro et al., 2013, 2021; Shiozaki et al., 2016; Tolar et al., 2016; Wan et al., 2018, 2021), and oceanographic cruises AR16 (https://www.bco-dmo.org/deployment/747056), JC156, and JC150. Coincident $NO_2^-$ and $NO_3^-$ (μM) were used to compute $NH_4^+$:DIN ratios. If coincident measurements of $NO_2^-$ were not available, then $NH_4^+$:DIN ratios were calculated with only $NO_3^-$. If $NO_3^-$ measurements were not made alongside $NH_4^+$, then $NO_3^-$ concentrations were extracted from the World Ocean Atlas 2018 (Garcia et al., 2019) monthly climatology at the closest grid cell. These data are available in Data Set S1.

Measured ammonia oxidation rates (N=696; nM day$^{-1}$) were also used for model-data assessment and showed an acceleration of rates from oligotrophic to eutrophic regions in agreement with the model (Fig. S3). Data were collated from published work (Clark et al., 2021; Dore and Karl, 1996; Mdutyana et al., 2020; Newell et al., 2013; Raes et al., 2020; Raimbault et al., 1999; Santoro et al., 2013, 2021; Shiozaki et al., 2016; Tolar et al., 2016; Wan et al., 2018, 2021) and are available in Data Set S2.

Measurements of $NH_4^+$ and $NO_3^-$ concentrations (μM) alongside $NH_4^+$- and $NO_3^-$-fueled primary production (μmol m$^{-3}$ day$^{-1}$) were used to determine the relationship between $NH_4^+$:DIN ratios and the proportion of net primary production that is fuelled by $NH_4^+$. While coincident measurements of these properties are not common, we compiled data from nine studies (Fernández et al., 2009; Joubert et al., 2011; Mdutyana et al., 2020; Metzler et al., 1997; Philibert, 2015; Rees et al., 2006; Thomalla et al., 2011; Wan et al., 2018; Yingling et al., 2021) providing 190 data points that together encompassed oligotrophic to eutrophic conditions from the tropics to the Southern Ocean. Measurements from the Gulf of Mexico (Yingling et al., 2021) were unique in that nutrient concentrations and uptake rates were not measured at precisely the same depths or stations. Coincident values were determined by calculating trends in depth via linear interpolation (Fig. S4). These data are available in Data Set S3.

Ammonia oxidation rates data from experiments involving pH changes were acquired directly from the papers presenting the results (Beman et al., 2011; Huesemann et al., 2002; Kitidis et al., 2011) by extraction from the text (where values were given) and from figures using the WebPlotDigitizer tool (https://automeris.io/WebPlotDigitizer/). Changes in ammonia oxidation rates were normalized to a pH of 8 (Fig. S1). These data are available in Data Set S4.

## 2.4 Phytoplankton relative abundance data

*Tara* Oceans expeditions between 2009 and 2013 performed a worldwide sampling of plankton in the upper layers of the ocean (Pierella Karlusich et al., 2020). We mined the 18S rRNA gene (V9 region) metabarcoding data set (Ibarbalz et al., 2019; de Vargas et al., 2015) by retrieving the operational taxonomic units (OTUs) assigned to eukaryotic phytoplankton from samples obtained from 144 stations (https://zenodo.org/record/3768510#.Xraby6gzY2w). Barcodes with greater than 85 % identity to phytoplankton sequences in reference databases were selected. The total diatom barcode reads in each sample was normalized

to the barcode read abundance of eukaryotic phytoplankton. We exclusively used the data sets corresponding to surface samples (5-9 m depth) because of greater sampling coverage in the *Tara* Oceans dataset, which accesses a broad range of $NH_4^+$:DIN ratios spanning many ocean biomes/provinces.

In addition, we analyzed the metagenomic read abundances for the single-copy photosynthetic gene *psbO*, an approach that

covers both cyanobacteria and eukaryotic phytoplankton and provides a more robust picture of phytoplankton cell abundances than rRNA gene methods (Pierella Karlusich et al., 2023). We retrieved the abundance tables from samples obtained from 145 stations (https://www.ebi.ac.uk/biostudies/studies/S-BSST761).

## 2.5 Statistical analyses

We explored the environmental drivers of change in phytoplankton relative abundance data (provided by *Tara* Oceans) with generalized additive models (GAMs) using the *mgcv* package in R (Wood, 2006) structured as:

$$Y = \alpha + s_1(x_1) + s_2(x_2) + \cdots + s_n(x_n) + \varepsilon, \tag{1}$$

Where $Y$ is the predicted response, $\alpha$ is the intercept, $s_n(x_n)$ represents a smooth function (specifically the $n^{th}$ thin-plate spline) fitted to the $n^{th}$ predictor variable $x_n$, and $\varepsilon$ is the model error. Thin-plate splines are flexible and widely used as a smoothing

method within GAMs that allow for non-linear relationships between predictors and response variables and do not require specificity around a functional form. They are well suited to handling ecological data where relationships are often non-linear and non-parametric. Predictor variables were mixed-layer depth (m), phosphate (μM), silicate (μM), dissolved iron (μM), and the $NH_4^+$:DIN ratio. Mixed layer depth, phosphate and silicate was measured *in situ* at the sample locations of Tara Oceans, while dissolved iron and $NH_4^+$:DIN ratios were provided by the model at the same location and month of sampling, since

measurements of these properties are scarce. In addition, phosphate and silicate concentrations were available as interpolated products from the World Ocean Atlas (Garcia et al., 2019). An alternative estimate of $NH_4^+$:DIN ratios was provided by the Darwin model (Follows et al., 2007). Predictor variables from models and World Ocean Atlas were extracted at the locations and months of sampling and different combinations of *in situ* and modelled variables were used to build GAMs. Mixed-layer depth, nutrients (phosphate, silicate and $NH_4^+$:DIN) and the relative abundance of phytoplankton taxa were $log_{10}$-transformed

prior to model building to ensure homogeneity of variance.

Before model testing, we calculated the variance inflation factors (VIFs) of independent variables to avoid multi-collinearity. All covariate VIFs were < 3, which indicates minimal multicollinearity. GAMs were computed using a low spline complexity (k = 3) that prevented overfitting and constrained the smooth functions represent only broad-scale trends in the data. We fit GAMs using all predictors (full model), then assessed the deviance explained by each predictor by fitting subsequent GAMS with each predictor in isolation, and by removing the predictor in question from the full model. The significance of a predictor was assessed by applying a smoothing penalty to only that predictor in the full model. Diagnostic plots were assessed visually, and predictive capacity was assessed via the percent of deviance explained by the model.

A two-sided Mann-Whitney U test was used to test for differences between the two distributions of diatom relative abundance separated by $NH_4^+$:DIN ratios < 4% and > 4%. The 4% threshold was used because it split the dataset in half and aligned with the point at which primary production transitioned from mostly new ($NO_3^-$-fueled) to regenerated ($NH_4^+$-fueled). This non-parametric test (performed with the *scipy* package in python) returned highly significant two-sided p-values ($p < 0.0001$) as indicated by ***.

## 3 Results and Discussion

### 3.1 Assessment of modelled $NH_4^+$ and $NH_4^+$:DIN

Concentrations of 0.1 μM $NH_4^+$ or greater exist over continental shelves and in regions of strong mixing with high rates of primary production and subsequent heterotrophy. This accumulation of $NH_4^+$ in productive regions is reproduced by our model (Fig. 3a). In these eutrophic systems, high $NH_4^+$ co-occurs with high $NO_3^-$ concentrations, so $NH_4^+$ makes a small contribution to total DIN (Fig. 3b). These regions include the eastern tropical Pacific, eastern boundary upwelling systems, the northwest Indian Ocean, the subpolar gyres and the Southern Ocean (although we note that the model underestimates $NH_4^+$ concentrations in the Southern Ocean). In contrast, low $NH_4^+$ concentrations of less than 0.05 μM pervade the oligotrophic gyres of the lower latitudes. As these regions also display very low $NO_3^-$ concentrations, $NH_4^+$ makes up a much higher fraction of total DIN in both the observations and our model, with the $NH_4^+$ peak occurring deeper in the water column (Fig. S2).

Eutrophic upwelling systems and oligotrophic waters differed in the major sinks of $NH_4^+$ (Fig. 3c), consistent with available observations and constraints from theory. The major difference was that ammonia oxidation represented $49 \pm 29$ % (mean ± standard deviation) of $NH_4^+$ sinks in eutrophic waters (here defined by surface nitrate > 1 μM) but this dropped to $32 \pm 9$ % in oligotrophic systems, where assimilation of $NH_4^+$ became more important. Measured rates of ammonia oxidation showed a positive relationship with surface $NO_3^-$ concentrations and this was reproduced by the model (Fig. S3), indicating that ammonia oxidation was indeed a greater proportion of the overall $NH_4^+$ budget in eutrophic regions. In agreement, isotopic methods have shown that the bulk of nitrogen assimilated by phytoplankton in oligotrophic waters is recycled (Eppley and Peterson,

1979; Fawcett et al., 2011; Klawonn et al., 2019; Van Oostende et al., 2017; Wan et al., 2021), implying that most nitrogen cycling occurs without ammonia oxidation. Our model reproduces this feature of oligotrophic systems (Fig. 3c). Overall, the model shows good fidelity to the available observations of $NH_4^+$ concentrations, $NH_4^+$:DIN ratios, and rates of $NH_4^+$ cycling that we compiled for this study (Fig. 3; Fig. S2-S3). Meanwhile, nitrogen fixation and anammox had very minimal contributions to $NH_4^+$ budgets on the global scale.

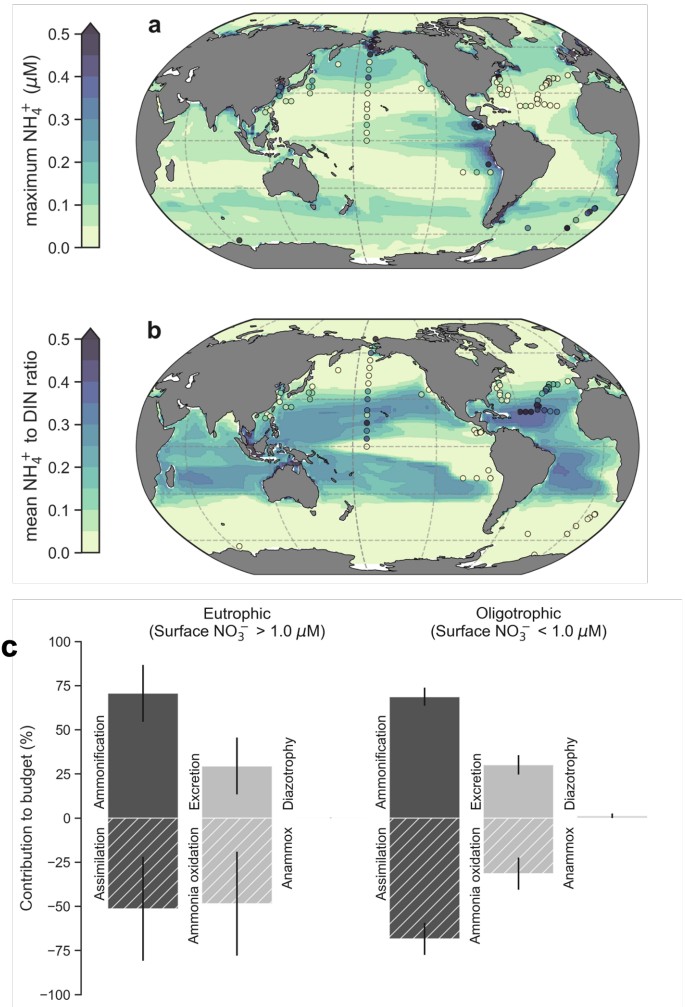

**Figure 3. Global patterns of $NH_4^+$ concentrations, its contribution to DIN, and $NH_4^+$ budgets within the photosynthetically active zone (phytoplankton biomass > 0.1 mmol C m⁻³). (a)** The simulated maximum $NH_4^+$ concentration. The maximum was chosen to emphasise basin-scale variations. **(b)** Average values of the $NH_4^+$:DIN ratio. Modelled values are annual averages of the preindustrial control simulation between years 2081-2100. Observed values following linear interpolation between the surface and 200 metres depth are overlaid as coloured markers. Only those profiles with at least 3 data points within the upper 200 metres are shown. **(c)** Global mean ± standard deviations of $NH_4^+$ fluxes separated into eutrophic and oligotrophic regions. Sources of $NH_4^+$ are represented by positive values and sinks by negative values.

## 3.2 Future enrichment of $NH_4^+$ in the ocean and its drivers

By the end of the 21st century (2081-2100), $NH_4^+$:DIN is projected to increase in over 98% of the photosynthetically active zone, where phytoplankton biomass exceeds 0.1 mmol C $m^{-3}$ (Fig. 4a). On average (± standard deviation), the fraction of DIN present as $NH_4^+$ increased by 6 ± 6 % from a preindustrial average of 11.5 ± 11.0 % to 17.5 ± 14 %, with enrichment exceeding 20% in regions with pronounced DIN gradients, such as at the boundary between eutrophic and oligotrophic regimes. The enrichment of $NH_4^+$ caused an expansion of regenerated production across the ocean, such that $NH_4^+$ overtook $NO_3^-$ as the main nitrogen substrate for phytoplankton growth in an additional 10% (73% to 83%) of the ocean's area. Regenerated production also increased as a proportion of net primary production from 60% to 63%. The greatest changes occurred within the 21st century (Fig. 4b), indicating a direct relationship between the severity of climate change and the magnitude of $NH_4^+$ enrichment within DIN.

Physical changes, a warming-induced stimulation of microbial metabolism, as well as ocean acidification all played a role in increasing $NH_4^+$:DIN. Among these factors, physical changes had the largest contribution, accounting for 55% of future trends (Fig. 4b). Physical changes decreased DIN to cause increases in $NH_4^+$:DIN in many regions (Fig 4c; Fig. S5) and occurred either through reduced physical supply (e.g., North Atlantic (Whitt and Jansen, 2020)) or increased demand and export of organic nitrogen in regions experiencing an increase in primary production due to losses in sea ice and increases in light (e.g., Arctic (Comeau et al., 2011)).

Ocean acidification, responsible for 25% of the $NH_4^+$:DIN increases, increased $NH_4^+$:DIN ubiquitously, but had the greatest effect in oligotrophic settings where DIN concentrations were lower, and minimal effects in eutrophic regions (Fig 4c; Fig. S5). We do note, however, that there is much uncertainty in the relationship between pH and ammonia oxidation rates (Bayer et al., 2016; Kitidis et al., 2011). To accommodate some of this uncertainty, we performed an idealized experiment with a weaker relationship between pH and ammonia oxidation that still fit the measurements well but that enforced a 10% decline in ammonia oxidation per 0.1 pH decline rather than 20% (Fig. S6). This reduced the influence of acidification by 10% or more and increased the contribution of the other stressors (Fig. S6). The effect of pH decline was, however, only influential to $NH_4^+$:DIN ratios in the subtropical gyres where $NH_4^+$:DIN ratios were already high. Thus, whether pH declines have a strong or weak effect on ammonia oxidation did little to change $NH_4^+$:DIN ratios in eutrophic regions where $NO_3^-$ is abundant and where diatoms represent a larger proportion of the phytoplankton community, and where coincidentally, shifts from low to higher $NH_4^+$:DIN would have the greatest impact on community composition.

Warming stimulated the nutrient demand of phytoplankton, which reduced DIN, a mechanism consistent with the effects of temperature on marine microbial recycling (Cherabier and Ferrière, 2022). While its global contribution was small at 13% (Fig. 4b), the stimulation of microbial metabolism had important effects at the boundaries of $NO_3^-$-rich regions by contracting

their areal extent, turning previously NO$_3^-$-rich waters to NO$_3^-$-poor waters (Fig. 4c; Fig. S5). Altogether, the individual contributions of physical change, acidification and stimulated metabolism diagnosed via our sensitivity experiments explained 93% of the full change in NH$_4^+$:DIN, indicating that the different drivers had small interactive effects that drove NH$_4^+$:DIN only slightly higher than their linear combination.


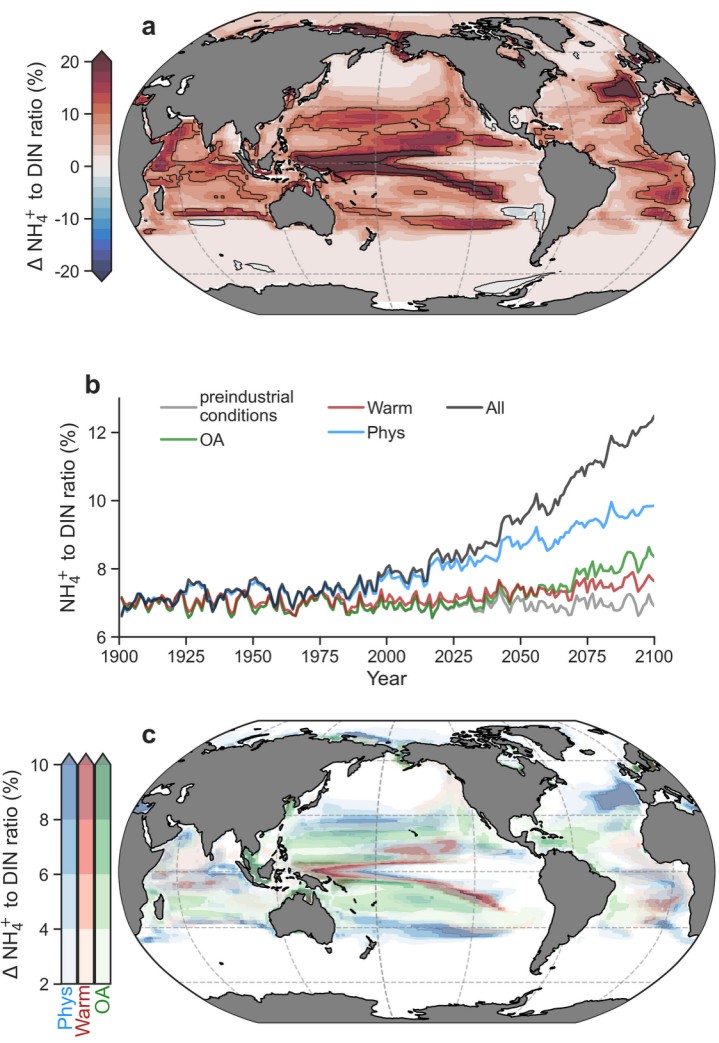

**Figure 4. Anthropogenic impacts on the NH$_4^+$ to DIN ratio in a high-emissions scenario within the photosynthetically active zone.** (a), The difference in the average NH$_4^+$ to DIN ratio at the end of the 21$^{st}$ century (2081-2100) with all anthropogenic impacts. (b), Global mean trends in the average NH$_4^+$ to DIN ratio in the different experiments: preindustrial control (grey), ocean acidification (OA; green), warming on metabolic rates (Warm; red), physical changes (Phys; blue) and all effects (All; black) according to the RCP8.5 climate change scenario. (c), Increases in the average NH$_4^+$ to DIN ratio due to physical changes (blue), effect of warming on metabolic rates (red) and ocean acidification on ammonia oxidation (green) from a multiple stressor perspective.


### 3.3 Simulated impacts on phytoplankton community composition

Our climate change simulations projected a future decline in the relative abundance of diatoms globally by an average of 3%, while local declines in the subantarctic, tropical, North Atlantic, North Pacific and Arctic Oceans sometimes exceeded 20% (Fig. 5a,b; Fig. S7). Our sensitivity experiments enabled an attribution of the major drivers, at least in a coarse-grained sense. At a global scale, the loss of diatom representation within marine communities in our model was driven by a combination of stimulated microbial metabolism (60% of full response in experiment "All") and physical changes (40% of full response in

experiment "All"), while ocean acidification had negligible effects (Figure 5c; Fig. S7). Ocean acidification had negligible effects because it largely raised $NH_4^+$:DIN ratios in oligotrophic subtropical gyres where diatoms were already of low proportion (Fig. 4c; Fig. S5). Averaged across the low latitude ocean (40ºS – 40ºN), diatoms also declined by an average of 3% driven by the same factors (60% microbial metabolism and 40% physical changes), while more dramatic but very regional declines of diatoms near or exceeding 20% were due primarily to physical changes (Fig. S7). These global and regional declines

have been predicted previously and are widely accepted to be due to a decline in bulk nutrient availability in the upper ocean (Bopp et al., 2005), although the large effect of stimulated metabolism here suggests that top-down grazing pressure, which is accelerated by warming, may also play a role (Chen et al., 2012; Rohr et al., 2023). That said, stimulating metabolism also increases phytoplankton nutrient demand, which eventually leads to greater DIN limitation (Cherabier and Ferrière, 2022). We indeed appreciate that the reduction of diatoms from phytoplankton communities as simulated by models is due to nutrient

losses, in particular declines in $NO_3^-$ (Kwiatkowski et al., 2020), and our simulations here, at least indirectly, are no different, since both nanophytoplankton and diatom biomass declined.

However, explicitly representing competition for $NH_4^+$ can provide a more nuanced view of why a decline in $NO_3^-$ might cause a decline in diatom relative abundance or shifts in any phytoplankton taxa for that matter. We cast this view specifically in

terms of an increase in competition for $NH_4^+$, and base this on two lines of evidence. First, a decline in the standing stock of DIN does not mean a decrease in its supply. In fact, rapid rates of primary production are measured in nutrient poor waters, which implies rapid recycling and thus a rapid resupply of DIN in the form of $NH_4^+$ (Bender and Jönsson, 2016; Matsumoto et al., 2016; Rii et al., 2016; Yang et al., 2019). This is akin to the bathtub analogy, where different volumes (i.e, nutrient concentrations) can result by varying the inflow (i.e., recycling) even when the outflow is constant (productivity). Second, we

take at face value the lower measured $NH_4^+$ affinities of diatoms compared with other phytoplankton (Litchman, 2007; Litchman et al., 2007), and we account for this competitive disadvantage explicitly in our ocean biogeochemical model (Fig. 2). The combination of intense competition for rapidly supplied $NH_4^+$ and the poor competitive ability of diatoms for $NH_4^+$ suggests that when $NO_3^-$ concentrations decline, competition for $NH_4^+$ increases, and declines in diatom relative abundance follow.


We recognize that other influential bottom-up and top-down stressors, such as growth limitation by other nutrients (Taucher et al., 2022), including $NO_3^-$ (Fig. 2), shifts in the light environment, and/or grazing pressure, which is also temperature dependent, are also influential to structuring phytoplankton communities (Brun et al., 2015; Margalef, 1978; Taucher et al., 2022). The fact that a warming-induced stimulation of metabolism was linked to 60% of the global mean diatom declines, for instance, could be due to a wide array of factors, not just the resulting increase in $NH_4^+$:DIN. Furthermore, we acknowledge that if a negative correlation between $NH_4^+$:DIN and diatom relative abundance exists, in our model or any observations, that this negative correlation may be confounded by covariates. If other factors are covarying with the $NH_4^+$:DIN ratio but are more influential to diatom relative abundance, this may lead to the erroneous attribution of a causative relationship between diatom relative abundance and $NH_4^+$:DIN ratios (i.e., a false positive).

Removing diatoms competitive disadvantage for $NH_4^+$ (i.e., equally competitive for $NH_4^+$) in our experiments with "model$_{compete}$" (see section 2.2.2 in the methods) mitigated the losses of diatom representation within future phytoplankton communities by 70% compared to the full response in the "All" experiment with model$_{control}$ (Fig. 5d-f). Losses in $NO_3^-$ still occurred in these experiments, and these losses in $NO_3^-$ caused declines in phytoplankton productivity and biomass, including both nanophytoplankton and diatoms everywhere outside of the polar regions (Fig. S7-S8). In the default model (model$_{control}$) diatoms experienced greater declines than nanophytoplankton, causing declines in their relative abundance. Importantly though, the global mean decline in diatom relative abundance in model$_{compete}$ was only 0.9% by 2081-2100 compared to 3% in model$_{control}$ (Fig. 5c,f). Physical changes, while important regionally, no longer exerted a global negative effect on their total nor relative abundance (blue line in Fig. 5f), while the negative effect of elevated microbial metabolism on relative abundance was ameliorated by 25% (Fig. 5f; Fig. S7-S8). In some areas diatoms even showed increased total and/or relative abundance where previously there were losses, including the Arctic, the tropical Pacific, the Arabian Sea, the North Atlantic, and the southern subtropics (Fig. 5d,e; Fig. S8). Outside of the Southern Ocean and the eastern boundary upwelling systems, physical changes that tended to reduce DIN concentrations now favoured diatoms, while elevated metabolism now had positive, rather than negative, effects in the tropical Pacific.

These experiments provide some potential insights into the factors controlling diatom niches in the global pelagic ocean. Regions where model$_{control}$ and model$_{compete}$ show similar changes are regions where other factors besides competition for $NH_4^+$ determine diatom competitiveness. A good example is the Southern Ocean, where iron, light and silicic acid are the major controls on diatom productivity and phytoplankton community composition (Boyd et al., 1999, 2000; Krumhardt et al., 2022; Llort et al., 2019). Accordingly, there is close correspondence in the model, evident by the matching outcomes of model$_{control}$ and model$_{compete}$. However, where model$_{control}$ and model$_{compete}$ predicted contrasting outcomes, the form of nitrogen, specifically $NH_4^+$:DIN and thus the intense competition for $NH_4^+$, exerted a dominant control.

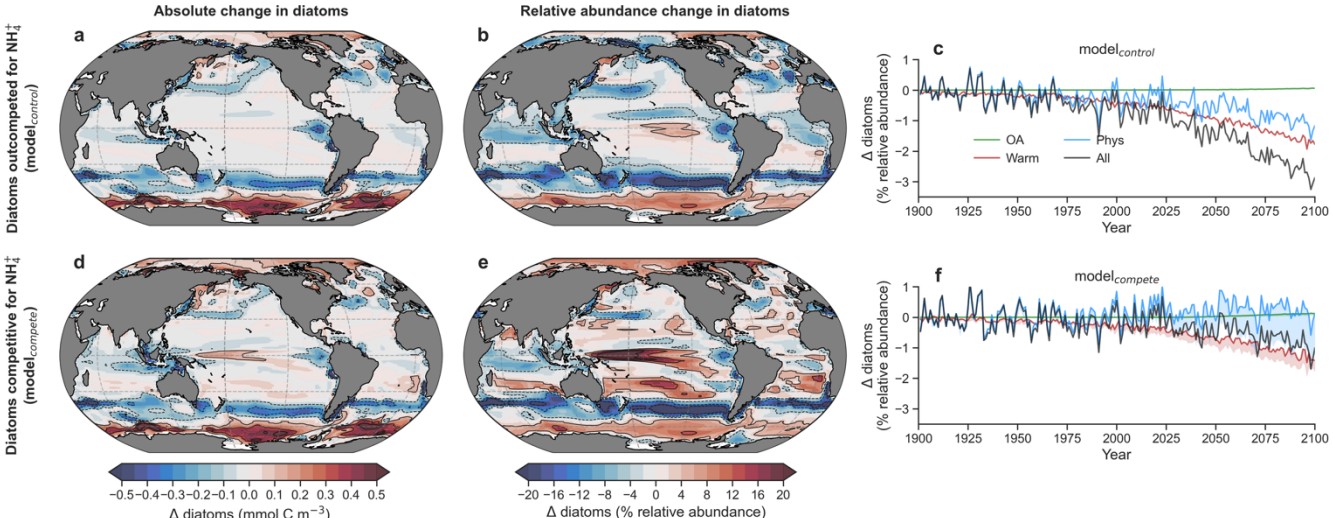

**Figure 5. Impact of NH4+ enrichment within DIN on diatoms.** (a), Mean change (Δ) in the absolute concentration of diatoms and (b) relative abundance of diatoms (%) by the end of the 21st century (2081-2100) as predicted by the control run of the ocean-biogeochemical model (model_control) under the RCP8.5 scenario and averaged over the photosynthetically active zone. (c), Global mean change in diatom relative abundance due to physical (circulation + light) changes (blue), warming effects on metabolic rates (red), ocean acidification effect on ammonia oxidation (green) and all stressors (black) for model_control. (d), The same as in (a), but for model_compete, where the NH4+ growth

limitation of the diatom PFT was made equal to the nanophytoplankton PFT. (e), The same as in (b), but for model_compete. (f), The same as in (c), but for model_compete. The shading shows the change between model_control and model_compete. Contours represent ± changes of 0.1, 0.3 and 0.5 mmol C m-3 and 4, 12 and 20 %.

## 3.4 Can we build confidence using observations?

So far, we have projected widespread increases in NH4+:DIN in a high-emissions scenario and determined that a large fraction of the projected declines in diatom relative abundance are due to their competitive exclusion by other phytoplankton in regions where NH4+ becomes more important as a nitrogen source. However, what do the observations tell us? Can this negative relationship between NH4+:DIN and diatom relative abundance be observed at a global scale? Evidence for local extirpation of diatoms by taxa more competitive for NH4+ has been reported by many studies (Andersen et al., 2020; Carter et al., 2005;

Donald et al., 2013; Glibert et al., 2016a; Örnólfsdóttir et al., 2004; Trommer et al., 2020), but is the relationship strong enough to play out across the wide biogeographic regimes in the ocean? Furthermore, is the parameterization of our model showcased in Figure 2 realistic? Does it reproduce observed shifts from new to regenerated production as NH4+ increases?

### 3.4.1 NH4+:DIN and regenerated production

We address the latter question first. If the model cannot reproduce observed shifts from NO3- to NH4+ fuelled primary production as NH4+:DIN changes, then we might be less confident in its projected increases in regenerated production, and by

extension less confident in the magnitude of projected declines in diatom relative abundance presented above. We collated parallel observations of $NH_4^+$:DIN ratios and rates of new and regenerated production from studies spanning tropical to polar environments (Fernández et al., 2009; Joubert et al., 2011; Mdutyana et al., 2020; Metzler et al., 1997; Philibert, 2015; Rees et al., 2006; Thomalla et al., 2011; Wan et al., 2018; Yingling et al., 2021). Such coincident measurements are rare. Nonetheless, this compilation was able to show the expected positive relationship between the $NH_4^+$:DIN ratio and the proportion of primary production that is regenerated (Fig. 6). While this relationship is expected, in that high $NH_4^+$ to DIN ratios should coincide with high rates of regenerated primary production, the functional form of this relationship is important yet not well known. The compilation of studies reveals that it is sharp and non-linear, and here we describe it using a fractional-order Monod function with an optimal half-saturation constant of $0.2 \pm 0.03$ µM/µM and an exponent of $0.5 \pm 0.05$ (Pearson's correlation = 0.69; $R^2$ (coefficient of determination) = 0.47; as compared to a linear relationship with an $R^2$ (coefficient of determination) = -1.13)). This quadratic function predicts that regenerated production contributes half of total net primary production when the standing stock of $NH_4^+$ is only $4 \pm 3$ % of total DIN. The data at hand therefore suggest that phytoplankton grow principally on $NH_4^+$ (regenerated production) and only transition to using $NO_3^-$ when $NH_4^+$ is substantially depleted to concentrations at or below 4% of total DIN.

A similarly sharp relationship emerged from our global ocean-biogeochemical model (Aumont et al., 2015) (grey dots in Fig. 6). This builds confidence in our modelled increases in regenerated production due to rising $NH_4^+$:DIN ratios, but why did the model behave similarly to the observed relationship? In the model, all phytoplankton are parameterized to have higher affinities for $NH_4^+$ over $NO_3^-$, consistent with laboratory studies (Litchman, 2007; Litchman et al., 2007). Their growth is supported by $NH_4^+$ only until $NO_3^-$ becomes sufficiently abundant to allow for higher growth rates (Fig. 2). In the model, this transition from $NH_4^+$ to $NO_3^-$ fuelled growth occurs at $NH_4^+$:DIN ratios of roughly 0.1 for the diatom functional type and roughly 0.025 for the nanophytoplankton function type under typical conditions. Hence, our model represents accelerated growth on $NO_3^-$ in both phytoplankton function types but only at very low $NH_4^+$:DIN ratios, and thus reproduces the sharp functional form that is observed.

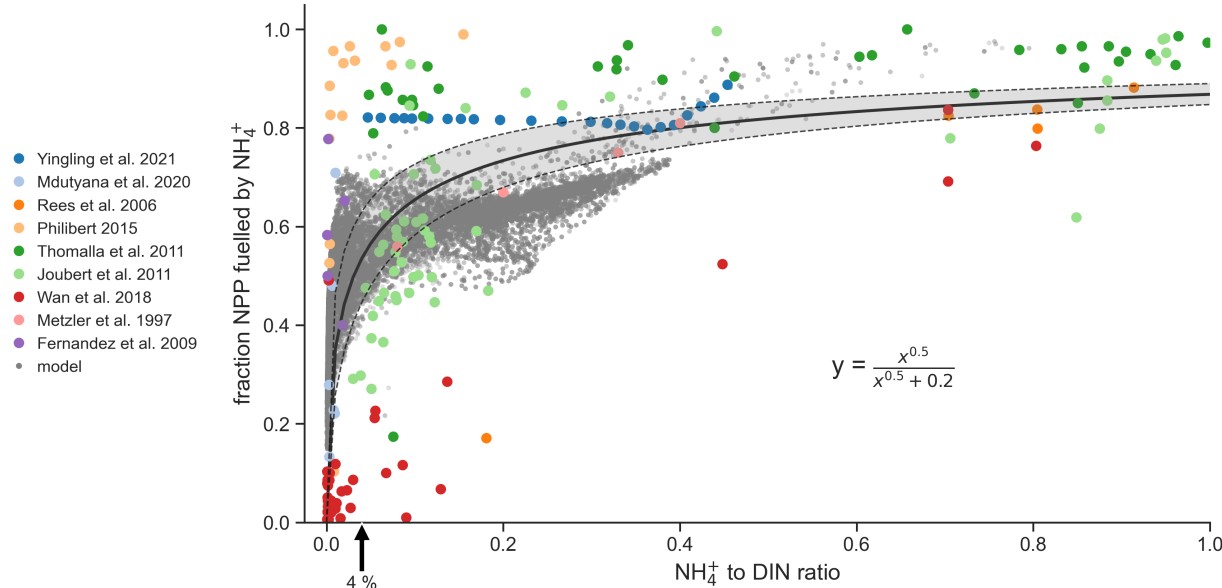

**Figure 6.** Coincident measurements of the $NH_4^+$ to DIN ratio and the fraction of net primary production (NPP) fuelled by $NH_4^+$ from nine studies (coloured dots) and as output by the ocean biogeochemical model run under preindustrial control conditions (grey dots). Black solid line is the best fit line to the observations and is described by the equation. Shading denotes one standard deviation.

### 3.4.2 $NH_4^+$:DIN and phytoplankton community composition

Next, we search for evidence of a relationship between $NH_4^+$:DIN ratios and phytoplankton community composition in the global ocean. While evidence from many localized studies in freshwater, brackish and marine environments suggests that increasing $NH_4^+$:DIN ratios have an effect on phytoplankton community composition, namely a negative effect on diatom relative abundance and a positive effect on cyanobacterial relative abundance (Berg et al., 2003; Carter et al., 2005; Donald et al., 2013; Fawcett et al., 2011; Klawonn et al., 2019; Van Oostende et al., 2017; Selph et al., 2021; Tungaraza et al., 2003; Wan et al., 2018), evidence for this relationship across the large-scale of the global ocean is lacking. We used two proxies of phytoplankton relative abundance from the *Tara* Oceans global survey, 18S rRNA gene metabarcodes (de Vargas et al., 2015) and *psbO* gene counts (Pierella Karlusich et al., 2023), combined with $NH_4^+$:DIN as predicted by our global ocean-biogeochemical model, to predict relative abundances of major phytoplankton taxa via Generalized Additive Models (GAMs; see Methods).

Our analysis revealed that what has been observed at local scales is apparent in the global *Tara* Oceans dataset. Essentially, elevated $NH_4^+$:DIN was consistently associated with declines in diatom relative abundance (Fig. 7a). The negative relationship between $NH_4^+$:DIN and diatom relative abundance was evident and significant in GAMs trained on both abundance proxies (18S rRNA and *psbO* gene counts), as well as when using different combinations of predictor variables: whether model-

derived, *in situ* measurements, interpolated products (Garcia et al., 2019), or even when switching out $NH_4^+$:DIN as predicted by our biogeochemical model with that provided by another (Follows et al., 2007) (Table S1). Importantly, the relationship between $NH_4^+$:DIN and diatom relative abundance remained consistently negative and significant despite the combination of predictor variables, which builds confidence in the statistical relationship. This was not the case for other predictors (phosphate, silicate, dissolved iron and mixed layer depth), which were prone to insignificance and/or sign changes depending on the combination of predictors used (Fig. S9-S13). $NH_4^+$:DIN also offered good explanatory power for diatom abundance compared to the other predictor variables, explaining between 18-30% of the deviance in the data for both 18S rRNA and *psbO* gene count data (Table S1).

We also saw some strong associations between $NH_4^+$:DIN and the relative abundance of dinoflagellates, *Prochlorococcus* and chlorophytes (Table S2; Fig. S14-S15). *Prochlorococcus* was positively related to $NH_4^+$:DIN, as expected, reflecting their superior affinity for $NH_4^+$ and dominance in oligotrophic gyres (Herrero et al., 2001; Litchman, 2007; Litchman et al., 2007; Matsumoto et al., 2016; Rii et al., 2016). The positive relationship between dinoflagellates and $NH_4^+$:DIN within eukaryotic phytoplankton, but not in the *psbO* gene counts, likely reflects the inclusion of non-photosynthetic (i.e., heterotrophic) dinoflagellate lineages with the 18S metabarcoding method that are excluded from the *psbO* method (Pierella Karlusich et al., 2023), and the proliferation of these types within systems enriched in reduced nitrogen (Glibert et al., 2016b). Like diatoms, chlorophytes were negatively related to $NH_4^+$:DIN. Interestingly, this is contrary to the outcomes of the freshwater studies that suggest a seasonal succession of increased chlorophyte concentrations as $NH_4^+$ concentrations increase following a diatom bloom on $NO_3^-$ (Andersen et al., 2020), as well as the high affinities that chlorophytes appear to have for $NH_4^+$ over $NO_3^-$ (Litchman, 2007; Litchman et al., 2007). However, the relative abundance of marine chlorophytes may also be affected by intense competition for $NH_4^+$ with cyanobacteria, which may have the competitive edge over small eukaryotes and push these taxa to niches with higher nutrient availability (Vannier et al., 2016). For chlorophytes, we therefore see a different relationship at the global scale compared to the local scale.

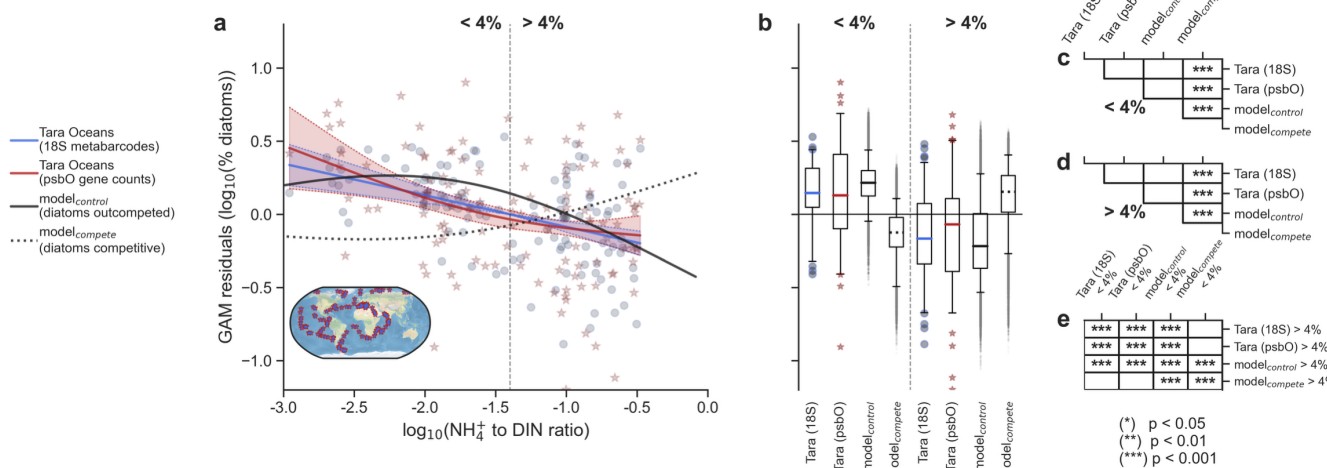

**Fig. 7. Effects of NH₄⁺ enrichment on diatom relative abundance.** (a), Partial dependence plot from the generalized additive model (GAM) showing the relationship between the NH₄⁺ to DIN ratio and the percent relative abundance of diatoms. When GAM residuals are positive this suggests that diatoms do better than predicted by a GAM without the NH₄⁺:DIN ratio as a predictor, and *vice versa*. Blue round markers and blue line fit are percent among eukaryotic phytoplankton (18S rRNA metabarcodes). Red star markers and red line fit are percent among all phytoplankton (*psbO* gene counts). Solid and dashed black lines are output from the ocean-biogeochemical model (N=16638) with and without competitive exclusion of diatoms for NH₄⁺. The vertical dotted line delineates when NH₄⁺ is 4% of DIN, which aligns with the point at which community primary production switches from predominantly NO₃⁻-fuelled to NH₄⁺-fuelled (Fig. 6). The inset map shows the locations of *Tara* Oceans samples (N=144). (b), Boxplots of the raw partial residuals from panel (a) but separated either side of the 4% NH₄⁺ to DIN threshold for percent among eukaryotic phytoplankton (blue), all phytoplankton (red), the ocean-biogeochemical model (solid black), and model without competitive exclusion of diatoms for NH₄⁺ (dashed black). Whiskers correspond to the 5ᵗʰ and 95ᵗʰ percentiles. Tables on the right denote significant pair-wise differences (Mann-Whitney U) amongst datasets when NH₄⁺:DIN is less than 4% (c), when it is more than 4% (d) and when comparing < 4% with > 4% datasets (e).

### 3.4.3 Building confidence in the model

To test whether the correct functional relationships emerge from our model, we performed the same GAM analysis that we performed in the previous section on diatom relative abundances predicted by our biogeochemical model. This model lacks a *Prochlorococcus* functional type and so does not allow us to comment on the relative abundance of this type but does ascribe its diatom functional type with a known competitive disadvantage for NH₄⁺ relative to the nanophytoplankton functional type (Fig. 2). This means that at high NH₄⁺:DIN ratios (low NO₃⁻) the nanophytoplankton type will always outcompete the diatom type.

As expected, the simulated diatom relative abundance was negatively related to NH₄⁺:DIN ratios (black line in Fig. 7a; deviance explained = 70%; p-value < 0.001). Interestingly, the relationship was also strongly non-linear and not dissimilar to that seen in the *Tara* Oceans data, with rapid losses of diatoms as NH₄⁺:DIN became greater than 4%. This threshold, where NH₄⁺ becomes 4% of total nitrogen stocks, aligned with the point at which primary production becomes dominated by

regenerated production (Fig. 6). This showcases (1) the intense recycling of $NH_4^+$ in the marine environment and competition for this coveted nutrient, (2) how diatoms are outcompeted as less primary production is fuelled by external $NO_3^-$ inputs, and

(3) how diatoms are major contributors to new primary production in the ocean (Fawcett et al., 2011). Additional statistical analysis showed that on either side of this 4% threshold the GAM predictions built from both the biogeochemical model and *Tara* Oceans data could not be statistically differentiated (Fig. 7b,c,d; Mann-Whitney U pair-wise tests). Both modelled and *Tara* Oceans data predicted similar values of diatom relative abundance within communities where $NH_4^+$:DIN was less than 4%, as well as in communities where $NH_4^+$:DIN was greater than 4% (Fig. 7b,e). Overall, the modelled and observed changes

in diatom relative abundance associated with $NH_4^+$:DIN appeared to be similar, at least statistically so. We stress that differences between the biogeochemical model and the *Tara* Oceans data no doubt exist. Nonetheless, the similarity between the model and the observations *may* mean that the negative relationship between $NH_4^+$:DIN and diatom relative abundance originate from the same mechanism, specifically being a competitive disadvantage of diatoms for $NH_4^+$.

### 3.4.4 The indirect effect of $NO_3^-$

We fully acknowledge that $NH_4^+$:DIN ratios covary strongly with $NO_3^-$ concentrations. Most of the projected increases in $NH_4^+$:DIN we report here are due to circulation changes that limit $NO_3^-$ injection from subsurface waters into surface waters (Fig. S5). Also, our GAM analysis of the *Tara* Oceans data could easily be replicated by replacing the $NH_4^+$:DIN ratio with $NO_3^-$ concentration as a key predictor. Indeed, this analysis showed similar results, with $NO_3^-$ being an equally strong predictor

of diatom relative abundance as $NH_4^+$:DIN. We therefore cannot discount a direct effect of $NO_3^-$ on diatom relative abundance in the *Tara* Oceans observations.

In our biogeochemical model, however, we can diagnose whether diatom relative abundance changes are directly due to competition for $NO_3^-$ or $NH_4^+$. This allows us to assess whether $NO_3^-$ concentration or the $NH_4^+$:DIN ratio are more appropriate

as a predictor of diatom relative abundance. The importance of $NH_4^+$ is exemplified by the fact that the negative relationship between $NH_4^+$:DIN and diatom relative abundance was reversed in model$_{compete}$ (black dotted line in Fig. 7a). Now positive rather than negative, this relationship differs statistically from that predicted from *Tara* Oceans data (Figure 7b-e).

This suggests that competition for $NH_4^+$ directly controls diatom relative abundance in our model. We fully acknowledge that

a scarcity of $NO_3^-$ is a major cause of $NH_4^+$ enrichment in our experiments because it drives competition for $NH_4^+$. However, we wish to emphasize that a potentially important mechanism of diatom decline in the community is due to their poor competitive ability for growth on $NH_4^+$, not directly because of decreases in $NO_3^-$. Decreases in $NO_3^-$ certainly affect diatom growth, but, in our model, they mostly do so indirectly by shifting the regime towards intense competition for $NH_4^+$. Given the statistical similarity between the *in situ* (*Tara* Oceans) and *in silico* (model$_{control}$) relationships (Fig. 7) and the dissimilarity

in model$_{compete}$, this points to $NH_4^+$:DIN as a key underlying driver of diatom relative abundance in the world ocean.

## 4 Conclusions

Here we have identified a potential enrichment of $NH_4^+$ in over 98% of the upper ocean (specifically the photosynthetically active zone) by the end of the 21st century under a high emissions scenario (Riahi et al., 2011). We expect that, given the evidence at hand, a widespread increase in $NH_4^+$-fuelled primary production and shifts in community composition, specifically some negative effects on the competitive niche of diatoms and any other taxa that could be considered $NO_3^-$ specialists and/or poor competitors for $NH_4^+$. These projections do not differ much from previous work (Bopp et al., 2005), but we recast the attribution of change in terms of competitive exclusion for $NH_4^+$, rather than bulk nutrient declines. In those places where nitrogen availability limits growth, diatoms suffer displacement by phytoplankton taxa with a greater affinity (i.e., competitive edge) for $NH_4^+$. The warming and physical changes that we simulate herein, and which drive $NH_4^+$ enrichment and diatom displacement, are expected (Bindoff et al., 2019), although the high-emissions scenario is now considered less likely than more moderate climate change scenarios. That said, we draw the link between the severity of climate change and the degree of $NH_4^+$ enrichment, such that our results can be scaled to consider more moderate scenarios. Also, the link between $NH_4^+$ enrichment and diatom displacement by more competitive phytoplankton has been demonstrated in numerous previous, albeit localized, studies, and here we demonstrate that it appears also on the global scale using the *Tara* Oceans dataset.

Fully elaborating on the link between environmental change and $NH_4^+$ enrichment also rests on many processes that are still not fully understood. For instance, an observed increase in summertime mixed layer depths may counter the effect of a strengthening pycnocline (Sallée et al., 2021) to increase $NO_3^-$ injection into the upper ocean as the ocean continues to respond to climate change. This might reduce competition for $NH_4^+$. Another good example is the incomplete understanding of the microbial loop and how it responds to environmental change. The microbial loop is driven by heterotrophic bacteria, which resupply $NH_4^+$ through mineralization of organic matter (Fig. 1). Increases in microbial metabolism were an important driver of the community shifts we projected. Yet, the representation in our model is simplistic. In fact, the microbial loop is not yet incorporated in detail within earth system models in general (Levine et al., 2025), but its response to warming can either elevate or depress regenerated production depending on assumptions made about bacterial physiology and function (Cherabier and Ferrière, 2022). The future balance of reduced ($NH_4^+$ and organic forms) to oxidized nitrogen and its impact on the state of marine ecosystems hinges on a suite of unexplored feedbacks between the marine microbial loop and environmental change. There is much work and research to be done in this space.

Many studies have identified that the open ocean habitat may be becoming more challenging for diatoms and more favourable for small eukaryotes and cyanobacteria. Reductions in $NO_3^-$ supply to the sunlit surface ocean have long been known as an important factor in the predicted loss of diatoms (Bopp et al., 2005). Meanwhile, iron stress appears to be growing in the diatom-dominated Southern Ocean (Ryan-Keogh et al., 2023) and fluctuates strongly across climatic modes of variability (Browning et al., 2023), silicic acid limitation is expected across the ocean in response to ocean acidification (Taucher et al.,

2022), and growing nitrogen limitation may make diatoms less adaptable as temperatures rise (Aranguren-Gassis et al., 2019). In this study, we add to these potential stressors of diatoms by highlighting the form of DIN. As before, $NO_3^-$ losses are important, but we emphasize that greater competition for $NH_4^+$ because of circulation changes and increased recycling, as well as the potential for a more nitrogen-limited Arctic, may further disadvantage diatoms and is expected to give cyanobacteria and other taxa with higher affinities for $NH_4^+$ a competitive edge. Furthermore, diatoms may be more susceptible to increases in competition for $NH_4^+$ in temperate waters, as cooler conditions appear to amplify their growth dependence on $NO_3^-$ (Glibert et al., 2016b; Parker and Armbrust, 2005), which is an additional mechanism not incorporated in this study. Notwithstanding the potential for evolution, these and other rapid changes may reduce diatom diversity (Lampe et al., 2018; Sugie et al., 2020), making diatoms susceptible to extirpation (Cael et al., 2021). If this is realized, ocean ecosystems look to shift towards longer, less productive food-chains underpinned by smaller, slower-growing phytoplankton (Sommer et al., 2002), with severe implications for the health of important fisheries and carbon storage. Further work is urgently needed to understand how the marine nitrogen cycle and key marine phytoplankton groups might respond to these growing challenges in an integrated manner.

**Code availability**

The model output and scripts to reproduce the analysis are available at https://doi.org/10.5281/zenodo.7630283. Developments to the PISCESv2 ocean-biogeochemical model code are freely available for download at https://github.com/pearseb/ORCA2_OFF_PISCESiso-N and is citable at https://doi.org/10.5281/zenodo.15612547.

**Data availability**

All data and materials used in the analysis are freely available. Nutrient data, nitrification rates, coincident nutrient concentrations with regenerated/new primary production rates, and ammonia oxidation rates relative to pH variations are provided in Supplementary Data 1-4. The biological data from the *Tara* Oceans sampling program are available at https://zenodo.org/record/3768510#.Xraby6gzY2w and https://ftp.ebi.ac.uk/biostudies/fire/S-BSST/761/S-BSST761/.

**Author contribution**

PJB conceptualized the study, curated the data, lead the analysis, investigation, software (code) development, ran model experiments, visualised the data and wrote the manuscript. JJPK and RET provided data and performed analysis, interpreted the results and contributed writing. RS provided data and visualisation, interpreted the results and edited the manuscript.

EMSW provided data, interpretation and edited the manuscript. CB and AT provided funding, computational resources, supervision, interpretation of the results and contributed to the writing and editing of the manuscript.

## Competing interest

The authors declare that they have no conflict of interest.

## Acknowledgements

Simulations and development were undertaken on Barkla, part of the High-Performance Computing facilities at the University of Liverpool. The authors wish to acknowledge use of the Ferret program (http://ferret.pmel.noaa.gov/Ferret/), climate data operators (https://code.mpimet.mpg.de/projects/cdo/), NetCDF Operators (http://nco.sourceforge.net/) and Python (www.python.org) for the analysis and graphics in this paper. Thanks to Xianhui Wan, Carolyn Buchwald and Alyson Santoro
who shared data, and ongoing discussions with Elena Litchman and Tyler Rohr.

## Financial support

PJB, RET and AT were supported by the ARISE project (NE/P006035/1), part of the Changing Arctic Ocean programme, jointly funded by the UKRI Natural Environmental Research Council (NERC) and the German Federal Ministry of Education and Research (BMBF). JJPK was supported by the Moore-Simons Project on the Origin of the Eukaryotic Cell, Simons
Foundation (735929LPI). EMSW and AT acknowledge support from the UKRI NERC grant NE/N009525/1, the Mid Atlantic Ridge project (FRidge). EMSW was also supported by the UKRI NERC grant NE/N001079/1 (Zinc, Iron and Phosphorus in the Atlantic). RET also acknowledges support from the UKRI NERC grant NE/W009536/1. CB acknowledges support from FEM (Fonds Francais pour l'Environnement Mondial), the French Government 'Investissements d'Avenir' programs OCEANOMICS (ANR-11-BTBR-0008), FRANCE GENOMIQUE (ANR-10-INBS-09-08), MEMO LIFE (ANR-10-LABX-
54), and PSL Research University (ANR-11-IDEX-0001-02), the European Research Council (ERC) under the European Union's Horizon 2020 research and innovation program (Diatomic; grant agreement No. 835067), and project AtlantECO.

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
