# Peer review of "Relative enrichment of ammonium and its impacts on open-ocean phytoplankton community composition under a high-emissions scenario"

_EGUsphere, 2024_

## Referee Comment (RC1)

**Review of Buchanan et al. (2025): "Oceanic enrichment of ammonium and its impacts on phytoplankton community composition under a high-emissions scenario"**

**Summary**

In their study, Buchanan and coauthors use an ocean biogeochemistry model forced by output from a climate model to investigate the impact of changes in nitrogen speciation on phytoplankton community structure. They find a shift towards more ammonium under a future climate, which is accompanied by a shift towards non-diatom phytoplankton and mostly driven by changes in ocean circulation.

Overall, the presentation of the results is clear, and this paper will be a valuable addition to literature assessing the response of phytoplankton to future climate change. Besides the new scientific findings, the authors compiled existing data related to nitrogen cycling from the literature for this paper; all these datasets are already made publicly available by the authors, which is valuable in itself. However, before publication of the manuscript, the presentation of the methods in the main text could be improved in my opinion, to provide readers with all information necessary to understand the results, i.e., by providing a more thorough description of how the model simulates the oceanic nitrogen cycle. Further, the description of the model sensitivity experiments should be revised to enhance clarity on what processes are (not) included in each experiment. I have no doubt that these issues can be addressed by the authors during the revisions, after which the study will be suitable for publication in Biogeosciences.

Please see the detailed explanation of all major and minor points below.

**Major comments**

My only somewhat major comment concerns the description of the methods, in particular the description of nitrogen cycling in the model:

The authors have made modifications to the model code describing nitrification, which is now a two-step process in their model. They currently fully describe these modifications in the supplementary information, but since any change done to the representation of the nitrogen cycle in their model is of relevance to the study at hand, I suggest adding the changes to the main text for full transparency. I acknowledge that the authors already briefly mention the updates in the main text (L. 96), but I would encourage the authors to reconsider the location of the detailed description of code changes, as bringing those to the main text would give any reader a much better overview of how nitrogen is cycled through their modeled ocean.

Further, I think it would help readers to see the modeled distributions of ammonium and its ratio to total dissolved inorganic nitrogen from the preindustrial control simulation in the main text. Especially in light of the changes the authors have made to the code, I think it would help readers, who aren't experts on modeling marine nitrogen cycling, to first demonstrate good performance of the new version of the model (by comparison with observations) and to show the baseline state in the main text before any sensitivity simulations or future projections are presented. Currently, the main text only shows relative changes for many properties (see Fig. 2, but also true for other figures), which makes it more difficult (than it has to be) for the reader to quickly evaluate what changes should be considered substantial.

In addition, I think it should be stated in more detail in the method section how biological nitrogen fixation is modeled, how atmospheric nitrogen deposition is treated, and how the two phytoplankton types differ in their affinity for different nitrogen species. Some of this information can be found throughout in the manuscript, but it would be more logical, at least for me, to have all this information presented in the method section for easier findability.

**Minor comments:**

L. 27: " a potentially underestimated"

L. 30: "fundamental  for the growth"

L. 67/68: Please add some references to support this statement: "numerous studies that showcase […]"

L. 100: How is biological nitrogen fixation parametrized?

L. 101: Can you elaborate on the atmospheric deposition of nitrogen? For example, how do you assume deposition to change in your future experiments?

L. 112: I suggest rephrasing to "forcing the *physical*-biogeochemical *ocean* model with monthly *atmospheric* output […]". Please state the variables you used from the IPSL model to force the ocean model. Also, ocean models are typically forced with atmospheric output at much higher frequency than monthly. Can you comment on why this choice was made and what impact you think this has on the results? What time step was used for the ocean model?

L. 122: For the sensitivity simulations, I suggest being more specific with respect to the model fields that were varied vs. held constant. For example, I assume you varied only all velocity components for the "Phys" experiment, but you could have equally varied only atmospheric wind variables. These two have different implications for what feedbacks are still possible in the ocean, and I therefore suggest stating this explicitly to avoid confusion. Similarly, for the "Warm" experiment, did you let temperature fields vary only within the biogeochemical subroutine to isolate the impact on biological rates or was it allowed to vary for the ocean model as a whole, i.e., thereby also affecting stratification and vertical nutrient supply? Or were atmospheric temperature fields varied? Lastly, for the "OA" experiment, did you hold pH fields constant (so that no feedbacks are possible due to, e.g., changing primary production) or did you vary atmospheric CO2 levels? I think it is important to be as specific as possible here and discuss the implications, so that the reader can better understand what processes are (not) captured in each experiment.

L. 129: This threshold reads somewhat arbitrary to me. Can you explain your reasoning, in particular why you didn't use a definition that I would consider more typical, i.e., defining the euphotic zone based on a light threshold (e.g., 1% of incoming PAR at the surface)? Are the two depths (very) different in your model?

L. 141/142: How often was this necessary? By neglecting nitrite in the equation, isn't the importance of ammonium inflated? Or did you in these cases also neglect nitrite from the model output? This is unclear. Please clarify and elaborate on the expected impacts on the evaluation and the importance of ammonium.

L. 146: In my opinion, "broad agreement" is a very subjective term. Can you provide a little more information in the main text and be more quantitative if possible (see also my general comment above)? Looking at Fig. S3, I am wondering why you didn't color the points for the model output. As it is, one can quickly see the agreement in magnitudes, but to assess spatial patterns more easily, using the same color code in both panels would help. Maybe reducing the number of colors/regions would help?

L. 179: Was the statistical model also built only with the data from the numerical model?

L. 183: I believe alpha is missing in the equation.

L. 188: Also from a PI-control simulation? Why was only the NH4:DIN ratio from a different model used but not the corresponding iron fields? Without having read the result section yet, I think the motivation for this choice (and its implications) could be better motivated in the method section.

L. 194: Please add an explanation what values below 3 mean.

L. 208-213: To streamline the text, I suggest deleting this part. This should be clear after the method section. Also, in section 2.2, there is no mention of sea ice. Can you check which text portion is correct?

L. 213-215: Please see my major comment above. I think you need to elaborate here and potentially move the evaluation figure to the main text. I think it would help readers who are not familiar with NH4 distributions to see the fields from Fig. S1 in the main text as a reference point for the change plots you show later in the paper.

L. 217: Can you explain what the "+- 6%" is? Spatial variability? Temporal variability?

L. 246: I am not convinced based on what you show that you can make this conclusion so easily. I realize that a quantitative attribution to the individual factors is complicated in fully-coupled simulations, but there might be important feedbacks and compensating effects between the factors that, by design, your relatively simple model setup cannot resolve. As such, I find your conclusion too strong here and suggest rewording. As a first-order test, have you run sensitivity experiments changing two out of the three factors instead of one at a time? This could help support your argument of linearity within the model framework of your study.

L. 258: What do present-day distributions of phytoplankton types look like in the model? How do they compare to observations?

Fig. 3: Please define the contours in the maps in the caption and/or directly in the figure. In addition, "model_control" and "model_compete" should be introduced in the method section.

L. 263: Following my above comment on the linearity, I suggest changing the language here. As it is currently written, this implies that the red, blue, and green curves in Fig. 3b perfectly add up to the black curve – I doubt this is the case.

 L. 269: Why "in some ways"?

L. 271: I am not sure I am fully up-to-date, but my impression is that many (most?) ocean biogeochemical models do differentiate between nitrate and ammonium (but not nitrite). I was first

wondering this here, but this might be something to mention in the introduction to provide a better context for the reader.

L. 280: In my opinion, this information should already be given in the method section (where you describe the model).

L. 286: Is zooplankton grazing temperature dependent in your model? The method section leaves it unclear which processes are included in the term "biological metabolism". I think it would be helpful to be more explicit about that (see also comment above).

L. 289: this

L. 296-304: To me, this belongs in the method section (see also comments above).

L. 307: I am not sure where to look to see the 70%. Is this a global integral? Can you clarify in the text?

L. 308: who

L. 365: Which model data are shown? From the preindustrial control?

L. 368: higher affinities than NO3? Please specify.

L. 388-393: Redundant with method section? Suggest deleting or shortening here to focus on the results.

Fig. 5: Could you comment on the difference in shape between the model-based fits in black and the Tara-based ones? As someone who is not too familiar with GAMs, the difference in the shape of the curves is what stands out to me. Looking at the underlying data, it seems like there are relatively fewer data points at the very low end of the x axis and around the 4% value – do you think there is room for data availability to impact the mismatch in the shape of the curves?

L. 463: You have not actually shown the circulation changes anywhere. For completeness, I suggest adding information on changes in the drivers in the supplement.

L. 493: albiet → albeit

L. 496: elaborating

L. 511: strong → strongly

Text S1: In Eq. 6, I believe it should say $[NO_2^-]$ → $[NO_3^-]$

---

## Author Comment (AC1)

**Review of Buchanan et al. (2025):**

*"Oceanic enrichment of ammonium and its impacts on phytoplankton community composition under a high-emissions scenario"*

**Summary**

In their study, Buchanan and coauthors use an ocean biogeochemistry model forced by output from a climate model to investigate the impact of changes in nitrogen speciation on phytoplankton community structure. They find a shift towards more ammonium under a future climate, which is accompanied by a shift towards non-diatom phytoplankton and mostly driven by changes in ocean circulation.

We would also hasten to add that the shift in phytoplankton community composition is also primarily driven by a metabolic stimulation by warming (see Fig 5).

Overall, the presentation of the results is clear, and this paper will be a valuable addition to literature assessing the response of phytoplankton to future climate change. Besides the new scientific findings, the authors compiled existing data related to nitrogen cycling from the literature for this paper; all these datasets are already made publicly available by the authors, which is valuable in itself.

We sincerely thank the reviewer for their careful and constructive feedback of our research.

However, before publication of the manuscript, the presentation of the methods in the main text could be improved in my opinion, to provide readers with all information necessary to understand the results, i.e., by providing a more thorough description of how the model simulates the oceanic nitrogen cycle. Further, the description of the model sensitivity experiments should be revised to enhance clarity on what processes are (not) included in each experiment.

We have acknowledged and addressed the reviewer's suggestions by incorporating more information in the methods section. This has involved the addition of a new methods section as well as more text in the description of our experiments.

I have no doubt that these issues can be addressed by the authors during the revisions, after which the study will be suitable for publication in *Biogeosciences*.

We again sincerely thank the reviewer for their positive recommendation.

Please see the detailed explanation of all major and minor points below.

**Major comments**

My only somewhat major comment concerns the description of the methods, in particular the description of nitrogen cycling in the model:

The authors have made modifications to the model code describing nitrification, which is now a two-step process in their model. They currently fully describe these modifications in the supplementary information, but since any change done to the representation of the nitrogen cycle in their model is of relevance to the study at hand, I suggest adding the changes to the main text for full transparency.

For brevity and readability, we originally opted to place this description in the supplement. As per the reviewers request, we added this information into the main text within the methods. However, on adding this information, we saw that this would be distracting to many readers of the paper because a full description of nitrification within the methods would dedicate 3 paragraphs and 8 equations. Adding a full description of nitrification was an important step in the technical developments needed to pursue this research, but the focus of this research is on diatoms response to increased $NH_4^+$ concentrations, which we show is driven by circulation and a warming-induced stimulation of phytoplankton metabolism, and nitrification therefore does not feature as a critical process, with the exception of its reponse to ocean acidification, but this effect is negligible on phytoplankton community composition (see Fig. 5).

However, the reviewer is right to want more clarity and descriptions of the model and the experiments. To accommodate the reviewer's request, we extend our initial description of the nitrogen cycle within the methods section but also maintain detailed information in the supplementary text for the interested reader. We also expand substantially on the N limitation parameterisation (see more in our answers below).

*Lines 103 – 131:*

*"2.1 The biogeochemical model*
*The biogeochemical model is the Pelagic Interactions Scheme for Carbon and Ecosystem Studies version 2 (PISCES-v2), which is detailed and assessed in Aumont et al. (2015). This model is embedded within version 4.0 of the Nucleus for European Modelling of the Ocean (NEMO-v4.0). We chose a 2° nominal horizontal resolution with 31 vertical levels with thicknesses ranging from 10 meters in the upper 100 meters to 500 meters below 2000 meters. Due to the curvilinear grid, horizontal resolution increases to 0.5° at the equator and to near 1° poleward of 50°N and 50°S.*

*We updated the standard PISCES-v2 (Aumont et al., 2015) for the purposes of this study, specifically by adding $NO_2^-$ as a new tracer. The PISCESv2 biogeochemical model already resolved the pools of $NH_4^+$, $NO_3^-$, dissolved oxygen, the carbon system, dissolved iron, phosphate, two kinds of phytoplankton biomass (nanophytoplankton and diatoms), two kinds of zooplankton biomass (micro- and meso-zooplankton), small and large pools of particulate organic matter, and dissolved organic matter (Aumont et al., 2015). While the model does not strictly represent picophytoplankton, implicit variations in the average cell size of the nanophytoplankton type affect nutrient uptake dynamics and may therefore encompass some functionality of picophytoplankton in oligotrophic systems (Aumont et al., 2015). The addition of $NO_2^-$ necessitated breaking full nitrification ($NH_4^+$ → $NO_3^-$) into its two steps of*

*ammonia ($NH_4^+$ ➔ $NO_2^-$) and nitrite oxidation ($NO_2^-$ ➔ $NO_3^-$). Both steps were simulated implicitly by multiplying a maximum growth rate by the concentration of substrate and limitation terms representing the effect of environmental conditions to return the realized rate. For ammonia oxidation, limitations due to substrate availability, light and pH determined the realised rate. For nitrite oxidation, limitations due to substrate availability and light affected the realised rate. All parameter choices were informed by field and laboratory studies and a detailed description is provided in the Supplementary Text S1.*

*New nitrogen is added to the ocean via biological nitrogen fixation, riverine fluxes, and atmospheric deposition. Nitrogen fixation and static riverine additions are equivalent to that presented in Aumont et al. (2015) and atmospheric deposition is maintained at preindustrial rates according to Hauglustaine et al. (2014) and applied as in Buchanan et al. (2021). Nitrogen is removed from the ocean via denitrification, anaerobic ammonium oxidation (anammox) and burial. The internal cycling of nitrogen involves assimilation by phytoplankton in particulate organic matter, grazing and excretion by zooplankton, solubilization of particulates to dissolved organics, ammonification of dissolved organic matter to $NH_4^+$, followed by nitrification of $NH_4^+$ and $NO_2^-$ via ammonia oxidation and nitrite oxidation (Supplementary Text S1)."*

I acknowledge that the authors already briefly mention the updates in the main text (L. 96), but I would encourage the authors to reconsider the location of the detailed description of code changes, as bringing those to the main text would give any reader a much better overview of how nitrogen is cycled through their modeled ocean.

We agree with the reviewer that a complete description of the oceanic nitrogen cycle within the main text would benefit the interested reader. However, for readability and to not distract from the main focus of the paper, which is about the response of the phytoplankton community composition to changes in the $NH_4^+$:DIN ratio, which itself is dependent on circulation changes and phytoplankton's response to warming (Fig. 5), we therefore opt to not distract too much from this narrative by focussing on details related to nitrogen fixation, denitrification, etc. We do, however, respectfully extend our simplified description of the nitrogen cycle (please see our response above) and provide an entirely new section within the methods that is dedicated to the nitrogen limitation routines of phytoplankton growth.

Further, I think it would help readers to see the modeled distributions of ammonium and its ratio to total dissolved inorganic nitrogen from the preindustrial control simulation in the main text. Especially in light of the changes the authors have made to the code, I think it would help readers, who aren't experts on modeling marine nitrogen cycling, to first demonstrate good performance of the new version of the model (by comparison with observations) and to show the baseline state in the main text before any sensitivity simulations or future projections are presented.

We fully agree with the reviewer and have brought the model assessment into the main text as the first results section. The new section is called:

*"3.1 Assessment of modelled $NH_4^+$ and $NH_4^+$:DIN"*

and reads as:

*Lines 345 – 373:*

*"Concentrations of 0.1 µM $NH_4^+$ or greater exist over continental shelves and in regions of strong mixing with high rates of primary production and subsequent heterotrophy. This accumulation of $NH_4^+$ in productive regions is reproduced by our model (Fig. 3a). In these eutrophic systems, high $NH_4^+$ co-occurs with high $NO_3^-$ concentrations, so $NH_4^+$ makes a small contribution to total DIN (Fig. 3b). These regions include the eastern tropical Pacific, eastern boundary upwelling systems, the northwest Indian Ocean, the subpolar gyres and the Southern Ocean (although we note that the model underestimates $NH_4^+$ concentrations in the Southern Ocean). In contrast, low $NH_4^+$ concentrations of less than 0.05 µM pervade the oligotrophic gyres of the lower latitudes. As these regions also display very low $NO_3^-$ concentrations, $NH_4^+$ makes up a much higher fraction of total DIN in both the observations and our model, with the $NH_4^+$ peak occurring deeper in the water column (Fig. S1).*

*Eutrophic upwelling systems and oligotrophic waters differed in the major sinks of $NH_4^+$ (Fig. 3c), consistent with available observations and constraints from theory. In eutrophic waters (here defined by surface nitrate > 1 µM), ammonia oxidation represented 49 ± 29 % (mean ± standard deviation) of $NH_4^+$ sinks, but this dropped to 32 ± 9 % in oligotrophic systems. Measured rates of ammonia oxidation showed a positive relationship with surface $NO_3^-$ concentrations and this was reproduced by the model (Fig. S2), indicating that ammonia oxidation was indeed a greater proportion of the overall $NH_4^+$ budget in eutrophic regions. In agreement, isotopic methods have shown that the bulk of nitrogen assimilated by phytoplankton in oligotrophic waters is recycled (Eppley and Peterson, 1979; Fawcett et al., 2011; Klawonn et al., 2019; Van Oostende et al., 2017; Wan et al., 2021), implying that most nitrogen cycling occurs without ammonia oxidation. Our model reproduces this feature of oligotrophic systems (Fig. 3c). Overall, the model shows good fidelity to the available observations of $NH_4^+$ concentrations, $NH_4^+$:DIN ratios, and rates of $NH_4^+$ cycling that we compiled for this study (Fig. 3; Fig. S2-S3)."*

[Figure]

Figure 3. Global patterns of NH₄⁺ concentrations, its contribution to DIN in the euphotic zone, and NH₄⁺ budgets. (a) The simulated maximum NH₄⁺ concentration within the euphotic zone. The maximum was chosen to emphasise basin-scale variations. (b) Average values of the NH₄⁺:DIN ratio. Modelled values are annual averages of the preindustrial control simulation between years 2081-2100. Observed values following linear interpolation between the surface and 200 metres depth are overlaid as coloured markers. Only those profiles with at least 3 data points within the upper 200 metres are shown. (c) Global mean ± standard deviations of NH₄⁺ fluxes separated into eutrophic and oligotrophic regions. Sources of NH₄⁺ are represented by positive values and sinks by negative values.

Currently, the main text only shows relative changes for many properties (see Fig. 2, but also true for other figures), which makes it more difficult (than it has to be) for the reader to quickly evaluate what changes should be considered substantial.

We agree with the reviewer that additional panels to our figures showing the absolute changes would be very helpful and informative. To address this, we have added Section 3.1 "Assessment of modelled NH₄⁺ and NH₄⁺:DIN", which now refers to Figure 3 that shows the maximum NH₄⁺ concentration and the mean NH₄⁺:DIN ratios in the global upper ocean. In particularly, the mean NH₄⁺:DIN ratios (Fig. 3b) can be easily compared with the change in NH₄⁺:DIN ratios (Fig. 4a).

We also expanded Figure 5 (previously Figure 3) to include two new panels that show the absolute changes in diatom concentrations and now reference these additional panels in the

text. The new figure 5 is:

[Figure]

**Figure 5.** Impact of $NH_4^+$ enrichment within DIN on diatoms. (a), Mean change ($\Delta$) in the absolute concentration of diatoms and (b) relative abundance of diatoms (%) by the end of the 21$^{st}$ century (2081-2100) as predicted by the control run of the ocean-biogeochemical model (model$_{control}$) under the RCP8.5 scenario and averaged over the euphotic zone. (c), Global mean change in diatom relative abundance due to physical (circulation + light) changes (blue), warming effects on metabolic rates (red), ocean acidification effect on ammonia oxidation (green) and all stressors (black) for model$_{control}$. (d), The same as in (a), but for model$_{compete}$, where the $NH_4^+$ growth limitation of the diatom PFT was made equal to the nanophytoplankton PFT. (e), The same as in (b), but for model$_{compete}$. (f), The same as in (c), but for model$_{compete}$. The shading shows the change between model$_{control}$ and model$_{compete}$.

In addition, I think it should be stated in more detail in the method section:

- How biological nitrogen fixation is modeled,
- How atmospheric nitrogen deposition is treated,
- How the two phytoplankton types differ in their affinity for different nitrogen species.

Some of this information can be found scattered throughout the manuscript, but it would be more logical to have it presented together in the method section for easier findability.

We have expanded on ammonia oxidation and nitrite oxidation in the methods (see response above and lines 110 – 131).

We have also added an entirely new section to the methods called "Isolating the effect of competition for $NH_4^+$", which reads as follows:

*Lines 189 – 245:*

[revised manuscript text omitted]

While we agree with the reviewer that a more complete description of the updates to the nitrogen cycle is important, we note that nitrogen fixation has not been altered from its default in this version of PISCESv2 and a description can be found at Aumont et al (2015). Nitrogen deposition, meanwhile, can be found at Buchanan et al. (2021) and we now point the reader to this paper. We also note that both nitrogen fixation and deposition have extremely small fluxes compared with primary production and nitrification, so their importance to the nitrogen cycle is negligible in the context of our study. Our quantification of the fluxes in the nitrogen cycle in what is now Figure 3c attests to this.

**Minor comments**

- **L. 27**: Typo - "an a potentially underestimated"
  Corrected.
- **L. 30**: Typo - "fundamental in for the growth"
  Corrected.
- **L. 67/68**: Please add references to support the statement "numerous studies that showcase [...]"
  Added.

- **L. 100**: How is biological nitrogen fixation parametrized?
  We point the interested reader to Aumont et al. (2015).
- **L. 101**: Please elaborate on atmospheric deposition of nitrogen — assumptions for future experiments?
  We have elaborated.
  *Lines 124-126:*
  *"New nitrogen is added to the ocean via biological nitrogen fixation, riverine fluxes, and atmospheric deposition. Nitrogen fixation and static riverine additions are equivalent to that presented in Aumont et al. (2015) and atmospheric deposition is maintained at preindustrial rates according to Hauglustaine et al. (2014) and applied as in Buchanan et al. (2021)."*
- **L. 112**: Suggest rephrasing and specifying which IPSL model output variables were used; justify why monthly forcing was used.
  Specified that the physical outputs used to force the BGC model are temperature, salinity, transports, short wave radiation and wind speeds.
- **L. 122**: Be specific about which fields were varied/held constant in sensitivity experiments.
  We have expanded to give more information.
  *Lines 168-177:*
  *"Experiment "Phys", for example, involved changing the ocean's circulation, temperature and salinity, and the resulting effects to light associated with sea-ice extent changes, but the ecosystem component of the model experienced only the preindustrial temperature, and atmospheric $CO_2$ was held at a preindustrial concentration of 284 ppm. In contrast, experiment "Warm" maintained the preindustrial climatological ocean state and atmospheric $CO_2$ at 284 ppm, but ensured that the ecosystem component saw increasing temperatures (T in °C) according to the RCP8.5 scenario, which scaled growth of phytoplankton types according to $1.066^T$ and heterotrophic activity (grazing and remineralisation) according to $1.079^T$ (Aumont et al., 2015). Experiment "OA" held the circulation and temperature effects on metabolism constant but involved the historical and future projected increase in atmospheric $CO_2$. This decreased pH and negatively affected rates of ammonia oxidation at a rate consistent with field measurements (Beman et al., 2011; Huesemann et al., 2002; Kitidis et al., 2011), specifically a loss of ~20% per 0.1 unit decrease in pH below 8.0 (Fig. S1)."*
- **L. 129**: Why define euphotic zone with a nutrient threshold rather than a light threshold? Clarify.
  Changed to *"upper ocean where primary production was active"*.
- **L. 141/142**: Clarify how ignoring nitrite in calculations impacts evaluation of ammonium.
  This will have negligible effects since $NO_2^-$ is of trace concentrations outside of the secondary nitrite maximum, which exists in low oxygen zones and beneath the euphotic zone.
- **L. 146**: Quantify what "broad agreement" means — subjective otherwise.
  *Lines 256-257: "Measured ammonia oxidation rates (N=696; nM day$^{-1}$) were also used for model-data assessment and showed an acceleration of rates from oligotrophic to eutrophic regions in agreement with the model (Fig. S3)."*
- **L. 179**: Clarify whether the statistical model was built only with model output.
  This is clarified in the text. We refer the reviewer to:
  *Lines 308-313:*
  *"Mixed layer depth, phosphate and silicate was measured in situ at the sample*

*locations of Tara Oceans, while dissolved iron and $NH_4^+$:DIN ratios were provided by the model at the same location and month of sampling, since measurements of these properties are scarce. In addition, phosphate and silicate concentrations were available as interpolated products from the World Ocean Atlas (Garcia et al., 2019). An alternative estimate of $NH_4^+$:DIN ratios was provided by the Darwin model (Follows et al., 2007). Predictor variables from models and World Ocean Atlas were extracted at the locations and months of sampling and different combinations of in situ and modelled variables were used to build GAMs."*

- **L. 183**: Alpha is missing in the equation.
  corrected.
- **L. 188**: Explain why NH4:DIN ratio is taken from another model but not iron fields.
  We note that any number of properties could be taken from any number of models. The focus of this study is on DIN, and PISCES is well known to perform well with dissolved Fe in the upper ocean, so we opt to not include additional representations of dissolved Fe.
- **L. 194**: Explain what values below 3 mean (context missing).
  Context added.
  *Line 332:*
  *"All covariate VIFs were < 3, which indicates minimal multicollinearity."*
- **L. 208–213**: Suggest removing redundant information.
  Removed.
- **L. 213–215**: Consider moving model evaluation figures into the main text.
  Moved what was supplementary Figure S1 to main text, which is now Figure 3.
- **L. 217**: Clarify what "+- 6%" refers to.
  standard deviation. Clarified in the text.
- **L. 246**: The conclusion about linearity of drivers seems too strong — consider rewording or running multi-driver sensitivity tests.
  Reworded.
  Lines 407 – 410:
  *"Altogether, the individual contributions of physical change, acidification and stimulated metabolism diagnosed via our sensitivity experiments explained 93% of the full change in $NH_4^+$:DIN, indicating that the different drivers had small interactive effects that drove $NH_4^+$:DIN only slightly higher than their linear combination."*
  Respectfully, the results do show that the linear combination amounts to 93% of the full response.
- **L. 258**: How does present-day phytoplankton distribution compare with observations?
  This is showcased and assessed in Aumont et al. (2015) for the PISCESv2 ocean biogeochemical model. We refer the interested reader to this publication for additional model details:
  *Lines 104-105:*
  *"The biogeochemical model is the Pelagic Interactions Scheme for Carbon and Ecosystem Studies version 2 (PISCES-v2), which is detailed and assessed in Aumont et al. (2015)."*
- **Fig. 3**: Define contours clearly in caption; introduce "model_control" and "model_compete" in methods.
  Complete.
- **L. 263**: Clarify language suggesting exact additive behavior of sensitivities.
  Clarified.
  Lines 479 – 482:
  *"Our sensitivity experiments enabled an attribution of the major drivers, at least in a*

*coarse-grained sense. At a global scale, the loss of diatom representation within marine communities in our model was driven by a combination of stimulated microbial metabolism (60% of full response in experiment "All") and physical changes (40% of full response in experiment "All"), while ocean acidification had negligible effects (Figure 5c; Fig. S7)."*

- **L. 269**: Clarify phrasing "in some ways."
  Changed to *"at least indirectly"*
- **L. 271**: Early note that most ocean models differentiate NO3 and NH4 (provides better context).
  We agree that many models do this, and that they are therefore well positioned to explore these competitive dynamics, but have yet to do so in a focussed way, which we have attempted herein.
  Lines 495 – 496:
  *"However, explicitly representing competition for $NH_4^+$ can provide a more nuanced view of why a decline in $NO_3^-$ might cause a decline in diatom relative abundance or shifts in any phytoplankton taxa for that matter."*

- **L. 280**: Some descriptions of model processes should move earlier (to methods).
  This has been moved earlier with a dedicated methods section.
- **L. 286**: Clarify whether zooplankton grazing is temperature-dependent.
  It is. Clarified.
- **L. 289**: Typo - "thids" should be corrected.
  Corrected.
- **L. 296–304**: Information could be moved to the method section.
- **L. 307**: Clarify what 70% refers to.
  Clarified.
- **L. 308**: Typo - "whom" usage check.
  Corrected.
- **L. 365**: Specify which model data shown (preindustrial control?).
  Clarified in the Figure legend.
- **L. 368**: Clarify "higher affinities than NO3."
  Clarifed.
- **L. 388–393**: Consider deleting redundant description.
  Deleted. We have removed this paragraph and extended the previous paragraph to be:

  *Lines 752 – 761:*
  *"Next, we search for evidence of a relationship between $NH_4^+$:DIN ratios and phytoplankton community composition in the global ocean. While evidence from many localized studies in freshwater, brackish and marine environments suggests that increasing $NH_4^+$:DIN ratios should have an effect on phytoplankton community composition, namely a negative effect on diatom relative abundance and a positive effect on cyanobacterial relative abundance (Berg et al., 2003; Carter et al., 2005; Donald et al., 2013; Fawcett et al., 2011; Klawonn et al., 2019; Van Oostende et al., 2017; Selph et al., 2021; Tungaraza et al., 2003; Wan et al., 2018), evidence for this relationship across the large-scale of the global ocean is lacking. We used two proxies of phytoplankton relative abundance from the Tara Oceans global survey, 18S rRNA gene metabarcodes (de Vargas et al., 2015) and psbO gene counts (Pierella Karlusich et al., 2023), combined with $NH_4^+$:DIN as predicted by our global ocean-biogeochemical model, to predict relative abundances of major phytoplankton taxa*

*via Generalized Additive Models (GAMs; see Methods)."*

- **Fig. 5**: Comment on curve differences between model and Tara dataset fits — could data sparsity cause shape mismatch?
  Yes it absolutely could. Tara Oceans has 144 data points to cover the entire globe, while the model on its native grid has 16638 surface grid cells, which is 115-fold more data points than the Tara Oceans dataset. This means that, simply due to data sparsity, different relationships may emerge.
  To accommodate this information, we have added to the Figure legend the number of data points for both the model and Tara Oceans data.
- **L. 463**: Circulation changes not shown; suggest adding supplemental info.
  Amended here to include a reference to the supplementary Figure 5.
- **L. 493**: Typo - "albiet" → "albeit"
  Corrected.
- **L. 496**: Typo - "elaboratuing"
  Corrected.
- **L. 511**: Typo - "strong" → "strongly"
  Corrected.
- **Text S1**: Equation typo — it should be [NO2^-] → [NO3^-].
  Corrected.

---

## Author Comment (AC2)

In this study, the authors apply a global marine biogeochemical model to investigate the changes in the relative abundance of diatoms in response to shifts in NH4:DIN ratio. This topic is of interest to both the modelling community and the broader research audience. Overall, the manuscript is well-written but would benefit from some structural reorganisation for readability. Additionally, several sections require further clarification and the inclusion of more supporting information.

We sincerely thank the reviewer for their positive view of our work and their thoughtful comments that have substantially improved the presentation of our work.

General comments

Typically, the presentation of model-data agreement (misfit) should precede the transient simulations, i.e., we should have "build confidence in the model" first. I suggest reorganising the discussion section so that the steady-state evaluation of the relationship between NH4:NO3 and diatom abundance appears first, followed by the transient simulations. Also, the "Model experiment" section should be moved after "Statistical analyses" in the methods.

We fully acknowledge the reviewer's suggestion and understand the motivation to rearrange the results.

In previous iterations of this manuscript, we structured the narrative to have the observed relationships first, followed by the modelling experiments. As such, our intuition was the same as the reviewer's. However, after many iterations and constructive feedback, we were encouraged to reorder the study and place the transient modelling experiments first. The motivation to place the modelling first is to present the experiments of "model$_{compete}$" as early as possible. We found that when these important results were placed at the end of the paper they were too easily lost. Meanwhile, presenting them earlier means that (1) they aren't lost and (2) the reader is aware that this study is primarily a model study.

We do, however, very much agree with the reviewer and reviewer 1 that some clarity could be gained by presenting some aspects much earlier and a model assessment up front. As such, we have included two entirely new sections: one in the methods, and another as the first results section. Please see our answers below for the revised text and figures for these new sections.

Several statistical techniques are applied in this study. I recommend providing more background information and justification for the selected values (e.g., VIFs and spline complexity). This would help readers unfamiliar with those tools to better understand the methodological choices.

We thank the reviewer for this suggestion and have included some additional information to aid the reader in why these choices were made. Under section 2.5:

Lines 300 - 310: *"We explored the environmental drivers of change in phytoplankton relative abundance data (provided by Tara Oceans) with generalized additive models (GAMs) using the mgcv package in R (Wood, 2006) structured as:*

$$Y = \alpha + s_1(x_1) + s_2(x_2) + \cdots + s_n(x_n) + \varepsilon ,$$
(6)

*Where $Y$ is the predicted response, $\alpha$ is the intercept, $s_n(x_n)$ represents a smooth function (specifically the $n^{th}$ thin-plate spline) fitted to the $n^{th}$ predictor variable $x_n$, and $\varepsilon$ is the model error. Thin-plate splines are flexible and widely used as a smoothing method within GAMs that allow for non-linear relationships between predictors and response variables and do not require specificity around a functional form. They are well suited to handling ecological data where relationships are often non-linear and non-parametric. Predictor variables were mixed-layer depth (m), phosphate (µM), silicate (µM), dissolved iron (µM), and the $NH_4^+$:DIN ratio. Mixed layer depth, phosphate and silicate was measured in situ at the sample locations of Tara Oceans, while dissolved iron and $NH_4^+$:DIN ratios were provided by the model at the same location and month of sampling, since measurements of these properties are scarce."*

Lines 331 - 333: *"Before model testing, we calculated the variance inflation factors (VIFs) of independent variables to avoid multi-collinearity. All covariate VIFs were < 3, which indicates minimal multicollinearity. GAMs were computed using a low spline complexity (k = 3) that prevented overfitting and constrained the smooth functions represent only broad-scale trends in the data."*

Although the biogeochemical model is based on a previously published version, this study applies a different nitrification configuration. The manuscript should provide at least a brief summary of how these changes affect key biogeochemical inventories (such as the relative abundance of the two phytoplankton types) and fluxes (including nitrogen fixation) to support the new model's validity.
We have provided a more extended description of the additions made to the PISCESv2 model in the methods, and provided an entirely new methods section that describes a key component of the nitrogen cycle: the nitrogen limitation parametersiation for diatoms and nanophytoplankton. We have also included a new section to the results that details a model-data assessment of N cycling in the upper ocean.

Lines 104 - 131: *"The biogeochemical model is the Pelagic Interactions Scheme for Carbon and Ecosystem Studies version 2 (PISCES-v2), which is detailed and assessed in Aumont et al. (2015). This model is embedded within version 4.0 of the Nucleus for European Modelling of the Ocean (NEMO-v4.0). We chose a 2° nominal horizontal resolution with 31 vertical levels with thicknesses ranging from 10 meters in the upper 100 meters to 500 meters below 2000 meters. Due to the curvilinear grid, horizontal resolution increases to 0.5° at the equator and to near 1° poleward of 50°N and 50°S.*

[revised manuscript text omitted]

Fig. 3 shows a 70% difference in the decline of delta % diatoms between the Model$_{control}$ and Model$_{compete}$. However, their delta µM C diatoms are very similar (Fig. S9). Could the changes in delta % diatoms during the transient simulation mainly result from differences in the initial conditions rather than the NH4:DIN ratio? If so, the decline in delta % diatoms might be primarily driven by a decrease in the overall nutrient pool rather than by competition with nanophytoplankton.

We appreciate the reviewer's question about this detail and we have improved our clarity in the methods. Our methods fully isolate the effect of competition for NH$_4^+$ on diatom relative abundance.

Lines 151 - 237:

[revised manuscript text omitted]

This brings another question to me. The manuscript does not discuss nanophytoplankton abundance during the transient simulations. Is the decline in their abundance really smaller than that of diatoms by the end of this century?

Yes this is correct. And that is why diatom relative abundance declines.

From Fig. S8a, the "delta other phytoplankton" are negative in most of the low latitude regions, where the NH4:DIN increases mostly. Since the title highlights impacts on phytoplankton community composition, I believe this is an important point. More discussion is needed on how both phytoplankton groups respond to the NH4:DIN shift.
We agree with the reviewer that this is an important point. As such, we discuss openly in the paper that the biomass and productivity of phytoplankton declines in general outside of the polar environments:

Lines 483 - 485:
*"We indeed appreciate that the reduction of diatoms from phytoplankton communities as simulated by models is due to nutrient losses, in particular declines in $NO_3^-$ (Kwiatkowski et al., 2020), and our simulations here, at least indirectly, are no different, since both nanophytoplankton and diatom biomass declined."*

Lines 560 - 563:
*"Losses in $NO_3^-$ still occurred in these experiments, and these losses in $NO_3^-$ caused declines in phytoplankton productivity and biomass, including both nanophytoplankton and diatoms everywhere outside of the polar regions (Fig. S7-S8). In the default model ($model_{control}$) diatoms experienced greater declines than nanophytoplankton, causing losses in relative abundance."*

Specific comments

Title: The majority of changes in diatom abundance due to changes in the NH4:NO3 ratio occur in trophic and subtropic regions (Fig. 3a and d), where NH4 concentration actually decreases (Fig. S6). Therefore, I suggest revising the phrase "enrichment of ammonium" in the title, as it may not accurately reflect the spatial trends shown in the results.
We have changed the title to avoid confusion from:
"Oceanic enrichment of ammonium and its impacts on phytoplankton community composition under a high-emissions scenario",
to:
"Relative enrichment of ammonium and its impacts on open-ocean phytoplankton community composition under a high-emissions scenario",

Line 27: an -> a
Corrected.

Line 30: remove the extra "in"
Corrected.

Fig. 1: Suggest adding labels to indicate which conditions are subject to anthropogenic pressure.

All processes are subject to anthropogenic climate change and resulting property changes. To make this more explicit, we have added a note to the figure legend:

*"All processes are affected by changes to seawater properties driven by large-scale climate change."*

Line 101: Are riverine inputs and nitrogen deposition influenced by anthropogenic forcing in the model?

Good question! No they are not and we have made this explicit in the methods:

*Lines 124 – 126:*

*"New nitrogen is added to the ocean via biological nitrogen fixation, riverine fluxes, and atmospheric deposition. Nitrogen fixation and static riverine additions are equivalent to that presented in Aumont et al. (2015) and atmospheric deposition is maintained at preindustrial rates according to Hauglustaine et al. (2014) and applied as in Buchanan et al. (2021)."*

Line 104-108: As mentioned in the major comments, please include some evaluations here, particularly for the nitrogen fixation since it's also affected by the forcings. Additionally, I couldn't find the information regarding the form of N introduced to the system through nitrogen fixation.

We have expanded our description of the nitrogen cycle within this version of PISCESv2 to address the reviewers request. This expansion reads as:

Lines 104 – 130:

*"The biogeochemical model is the Pelagic Interactions Scheme for Carbon and Ecosystem Studies version 2 (PISCES-v2), which is detailed and assessed in Aumont et al. (2015). This model is embedded within version 4.0 of the Nucleus for European Modelling of the Ocean (NEMO-v4.0). We chose a 2° nominal horizontal resolution with 31 vertical levels with thicknesses ranging from 10 meters in the upper 100 meters to 500 meters below 2000 meters. Due to the curvilinear grid, horizontal resolution increases to 0.5° at the equator and to near 1° poleward of 50°N and 50°S.*

*We updated the standard PISCES-v2 (Aumont et al., 2015) for the purposes of this study, specifically by adding $NO_2^-$ as a new tracer. The PISCESv2 biogeochemical model already resolved the pools of $NH_4^+$, $NO_3^-$, dissolved oxygen, the carbon system, dissolved iron, phosphate, two kinds of phytoplankton biomass (nanophytoplankton and diatoms), two kinds of zooplankton biomass (micro- and meso-zooplankton), small and large pools of particulate organic matter, and dissolved organic matter (Aumont et al., 2015). The addition of $NO_2^-$ necessitated breaking full nitrification ($NH_4^+ \rightarrow NO_3^-$) into its two steps of ammonia ($NH_4^+ \rightarrow NO_2^-$) and nitrite oxidation ($NO_2^- \rightarrow NO_3^-$). Both steps were simulated implicitly by multiplying a maximum growth rate by the concentration of substrate and limitation terms representing the effect of environmental conditions to return the realized rate. For ammonia oxidation, limitations due to substrate availability, light and pH determined the realised rate. For nitrite oxidation, limitations due to substrate availability and light affected the realised rate. All parameter choices were informed by field and laboratory studies and a detailed description is provided in the Supplementary Text S1.*

*New nitrogen is added to the ocean via biological nitrogen fixation, riverine fluxes, and atmospheric deposition. Nitrogen fixation and static riverine additions are equivalent to that presented in Aumont et al. (2015) and atmospheric deposition is maintained at preindustrial rates according to Hauglustaine et al. (2014) and applied as in Buchanan et al. (2021). Nitrogen is removed from the ocean via denitrification, anaerobic ammonium oxidation (anammox) and burial. The internal cycling of nitrogen involves assimilation by phytoplankton in particulate organic matter, grazing and excretion by zooplankton, solubilization of particulates to dissolved organics, ammonification of dissolved organic matter to $NH_4^+$, followed by nitrification of $NH_4^+$ and $NO_2^-$ via ammonia oxidation and nitrite oxidation (Supplementary Text S1). "*

With regard to the effect of nitrogen fixation, its contribution to nitrogen cycling is now showcased in what is now Figure 3c.

Line 121: Please provide a clearer description of the "changing circulation ('Phys')" configuration. For example, does it include stronger stratification? It is not clear which specific factors are incorporated under this forcing. Later in the text (Line 211), changes in sea-ice loss are also mentioned as part of this forcing, so clarification is needed regarding which processes are included.
To address the reviewers concern, we have expanded on this paragraph to explain more fully what the sensitivity experiments were composed of:

Lines 166 – 175:
*"Experiment "Phys", for example, involved changing the ocean's circulation, temperature and salinity, and the resulting effects to light associated with sea-ice extent changes, but the ecosystem component of the model experienced only the preindustrial temperature, and atmospheric $CO_2$ was held at a preindustrial concentration of 284 ppm. In contrast, experiment "Warm" maintained the preindustrial climatological ocean state and atmospheric $CO_2$ at 284 ppm, but ensured that the ecosystem component saw increasing temperatures (T in °C) according to the RCP8.5 scenario, which scaled growth of phytoplankton types according to $1.066^T$ and heterotrophic activity (grazing and remineralisation) according to $1.079^T$ (Aumont et al., 2015). Experiment "OA" held the circulation and temperature effects on metabolism constant but involved the historical and future projected increase in atmospheric $CO_2$. This decreased pH and negatively affected rates of ammonia oxidation at a rate consistent with field measurements (Beman et al., 2011; Huesemann et al., 2002; Kitidis et al., 2011), specifically a loss of ~20% per 0.1 unit decrease in pH below 8.0 (Fig. S1)."*

Line 129: Please provide a reference or justification for this criterion (0.1 mmol C m-3).
We have removed the reference to the "euphotic zone" here to avoid confusion. The sentence now reads:
Lines 181 – 185:
*"We calculated changes at each grid cell by averaging over the upper ocean where primary production was active, which was defined as those depths where total phytoplankton biomass was greater than 0.1 mmol C $m^{-3}$."*

Line 160-163: Why the rate saturates when pH > 8? Base on the equation in Fig. S5 the rate is supposed to keep increase.

We chose a pH value where pH is no longer limiting to the maximum possible rate of ammonia oxidation, but beneath which ammonia oxidation becomes limited by pH declines as more $NH_3$ is converted to $NH_4^+$. Figure S5 (now Fig. S1) clearly shows this cut off, but we acknowledge that this was not clear in the equation within Figure S5 (now Fig. S1). We have rectified this inconsistency by including a new equation to the figure. We also updated the equations in Fig. S6.

Equation (1): at least one item is missing before +s1(x1).

Corrected.

Line 183: the intercept $\alpha$ is missing in the equation.

Corrected.

Line 183: "thin-plate spline" is not a trivial term, it would be helpful if the author could provide a brief explanation or background information in the text.

We agree that this necessitates a bit more explanation:

Lines 295 – 299:

*"Where Y is the predicted response, $\alpha$ is the intercept, $s_n(x_n)$ represents a smooth function (specifically the $n^{th}$ thin-plate spline) fitted to the $n^{th}$ predictor variable $x_n$, and $\varepsilon$ is the model error. Thin-plate splines are flexible and widely used as a smoothing method within GAMs that allow for non-linear relationships between predictors and response variables and do not require specificity around a functional form. They are well suited to handling ecological data where relationships are often non-linear and non-parametric."*

Line 184: independent variable. -> independent variable x.

Corrected.

Line 193-194: Please provide a bit more information regarding the VIFs and the criterion.

Addressed.

*"All covariate VIFs were < 3, which indicates minimal multicollinearity."*

Line 210-211: The full name was mentioned in the Methods and it is sufficient using only RCP8.5 here.

This has been removed.

Line 211: "sea-ice loss" should be mentioned already in the methods instead of here.

This has been mentioned now on line 167. We thank the reviewer for catching this.

Line 217: What does the ±6 stands for?

standard deviation. Clarified in the text.

Line 219: Please give specific locations for examples for the "oceanographic fronts".
replaced on line 371 by *"as at the boundary between eutrophic and oligotrophic regimes"*

Line 225-242: When comparing Fig. 3b and 3e, the major contribution to the changes in diatom abundance appears to come from the "phys", which aligns with its 55% contribution to the NH4:NO3 ratio. However, although the contribution from OA (25%) is about double that of Warming (13%), OA appears to have almost no effect on the changes in diatom between Fig.3b and 3e. This discrepancy should be addressed and further explained in the manuscript.

We completely agree with the reviewer and we have allocated an explanatory sentence to this result:

Lines 472 – 476:

*"At a global scale, the loss of diatom representation within marine communities in our model was driven by a combination of stimulated microbial metabolism (60% of full response in experiment "All") and physical changes (40% of full response in experiment "All"), while ocean acidification had negligible effects (Figure 5c; Fig. S7). Ocean acidification had negligible effects because it largely raised $NH_4^+$:DIN ratios in oligotrophic subtropical gyres where diatoms were already of low proportion (Fig. 4c; Fig. S5)."*

Fig. 3 and Fig. S9:  The delta µM C diatoms of "All" are comparable between $Model_{control}$ and $Model_{compete}$ in Fig. S9, yet the delta % diatoms in Fig. 3 show much larger difference. Does this imply that the total diatom biomass is substantially higher in $Model_{compete}$? Additionally, in $Model_{control}$, the delta µM C diatoms decline under "Warm" is about 2.5 times greater than Phys (Fig. S9c), but their  delta % diatoms declines are similar. Does this indicate that the diatom biomass is much higher in "Phys"? These discrepancies should be clarified in the text, as they are important for interpreting the results.

The reviewer brings attention to an important nuance of the results: that both diatom and nanophytoplankton biomass decrease in the "All" experiment in both $model_{control}$ and $model_{compete}$ experiments, but that the diatom relative abundance decreased much less in $model_{compete}$. This means that diatoms still decrease in $model_{compete}$ and their losses were greater than nanophytoplankton losses, but to a much lesser extent than in $model_{control}$.

We have clarified this nuance in the text:

Lines 558 – 571:

*"Removing diatoms competitive disadvantage for $NH_4^+$ (i.e., equally competitive for $NH_4^+$) in our experiments with "$model_{compete}$" (see section 2.2.2 in the methods) mitigated the losses of diatom representation within future phytoplankton communities by 70% compared to the full response in the "All" experiment with $model_{control}$ (Fig. 5d-f). Losses in $NO_3^-$ still occurred in these experiments, and these losses in $NO_3^-$ caused declines in phytoplankton productivity and biomass, including both nanophytoplankton and diatoms everywhere outside of the polar regions (Fig. S7-S8). In the default model ($model_{control}$) diatoms experienced greater declines than nanophytoplankton, causing declines in their relative abundance. Importantly though, the global mean decline in diatom relative abundance in $model_{compete}$ was only 0.9% by 2081-2100 compared to 3% in $model_{control}$ (Fig. 5c,f). Physical changes, while important regionally,*

*no longer exerted a global negative effect on their total nor relative abundance (blue line in Fig. 5f), while the negative effect of elevated microbial metabolism on relative abundance was ameliorated by 25% (Fig. 5f; Fig. S7-S8). In some areas diatoms even showed increased total and/or relative abundance where previously there were losses, including the Arctic, the tropical Pacific, the Arabian Sea, the North Atlantic, and the southern subtropics (Fig. 5d,e; Fig. S8). Outside of the Southern Ocean and the eastern boundary upwelling systems, physical changes that tended to reduce DIN concentrations now favoured diatoms, while elevated metabolism now had positive, rather than negative, effects in the tropical Pacific."*

In relation to this comment by the reviewer: "Additionally, in Model$_{control}$, the delta µM C diatoms decline under "Warm" is about 2.5 times greater than Phys (Fig. S9c), but their delta % diatoms declines are similar. Does this indicate that the diatom biomass is much higher in "Phys"?" we stress that the changes in diatom relative abundance are not dependent on the changes in absolute PFT biomass, as it depends entirely on how much nanophytoplankton change in relation to diatoms. So in experiment "Warm", the declines in diatom and nanophytoplankton biomass are indeed stronger than in "Phys", but because in both cases the relative declines are similar, both "Phys" and "warm" have similar effects on the relative abundance of diatoms.

We feel that the revised text in the methods has clarified these points and resolved any prior ambiguities.

Line 289: thid -> the or this?
Corrected.

Line 296-304: move to method. Also, was the model$_{compete}$ simulation spun up before the transient simulation? Such information is missing in the text.
We agree! We have created a new methods section that is focussed on these experiments and also explains the nitrogen limitation function in PISCESv2. This reads as follows:

Lines 184 – 237:
*"2.2.2 Isolating the effect of competition for $NH_4^+$*

[revised manuscript text omitted]

Line 306: Why surprisingly? Is 70% too high or too low?
Removed.

Line 493: albiet -> albeit
Corrected.

Line 496: ellaboratiung -> elaborating
Corrected.

Fig. S3: If possible, please add one panel that displays model results from where observations exist.
Included in the figure.

Fig. S8: I believe the unit (or scale) for the right panels is wrong.
We have corrected the figure.

---

## Author Comment (AC3)

**Review of "Oceanic enrichment of ammonium and its impacts on phytoplankton community composition under a high-emissions scenario" by P. Buchanan et al. for Biogeosciences**

The manuscript by Buchanan and coauthors employs a physical biogeochemical model with an improved nitrogen (N) cycle representation to investigate the effect of climate change on the availability of different dissolved inorganic nitrogen (DIN) sources (mainly nitrate, NO3 and ammonium, NH4) to phytoplankton, and the consequences for phytoplankton diversity. They find that over most of the surface ocean the availability of NH4 increases relative to NO3, with a global mean ratio increasing from ~7% to 12% by the end of the century. The most significant changes are predicted in mid- to low-latitude regions. The model also projects a global decline in diatom biomass of about 3%.

We hasten to note that it is not diatom biomass that declines by 3%, but their relative abundance compared to nanophytoplankton.

By comparing model output with geochemical rate observations and analysis of Tara Oceans's genomic dataset, the authors suggest that this shift towards higher NH4/DIN ratio supports (1) an increase in regenerated production, and (2) a decrease in the relative abundance of diatoms, which are more dependent on NO3, in favor of smaller phytoplankton groups (pico- and nano-phytoplankton) that are more reliant on NH4.

As climate change reshapes the oceanic ecosystem, it is clear that there will be ecological winners and losers, but the outcomes remain highly uncertain, both in magnitude and patterns. Buchanan and coauthors approach this question from an interesting angle, focusing on shifts in the form of dissolved inorganic nitrogen and how these changes may affect phytoplankton diversity.

We sincerely thank the reviewer for their keen interest in the topic.

This is an interesting study that addresses a globally relevant topic through the use of a state-of-the-art model and a thoughtful analysis of observational data. The model projections and the analysis of the Tara dataset are stimulating and valuable, both on their own and when combined to support a mechanistic interpretation of the changes observed. In fact, the observational constraints presented here could easily become benchmark for future ocean biogeochemical models, particularly for evaluating their representation of DIN dynamics. For these reasons, I believe the study is appropriate for Biogeosciences, and I am ultimately supportive of publication.

We again sincerely thank the reviewer for their support of our work and the results presented herein.

However, the manuscript is dense with information, presenting several complex analyses and interpretations that are not always clearly or convincingly explained.

At times, I found myself wondering whether the results might be more effectively communicated if the study were divided into two separate papers: one focused on the present-day ocean and the role of NH4/DIN in shaping phytoplankton communities, and another dedicated to future projections and their mechanistic interpretation. The narrative structure of the paper feels somewhat meandering, moving from future model projections and sensitivities to observational analyses, then to model-observation comparisons. This steadily introduces new amounts of information, and a series of new questions are raised halfway through the result section (notably in Section 3.3), increasing the complexity of the narrative. As reflected in my detailed comments below, while I found the study rich with interesting results, I often struggled to follow the logic of the explanations and interpretations— challenges compounded by the paper's structure.

We are thankful for the reviewer's thoughtful comments on how the narrative should be presented. In our revised version, we have added new sections to the methods and results that improve the clarity of the paper and that give more details to the modelling approach. Please see our more detailed responses below.

Furthermore, I am unconvinced by the more assertive interpretation that the modeled diatom decline reflects more intense competition for NH4, rather than the more straightforward effect of a general decline in NO3 supply and concentrations—on which diatoms depend directly, given the model's DIN uptake formulation. To be fair, this interpretation is presented as a suggestion in multiple parts of the paper, and is not necessary for the paper to stand on its own, given the range of interesting results and analyses provided. For example, I find the phrasing of the abstract to be balanced, but many parts of the Results and Discussion present this idea with much less nuance. The authors themselves acknowledge at multiple points that it is difficult to disentangle the effects of an increase in NH4/DIN from those of a decrease in NO3 concentrations, and in my view there is no contradiction in proposing that diatom declines reflect both effects. I'm not sure one can simply isolate competition for NH4 as the main driver of the changes observed — especially given how important circulation driven changes are on a point by point basis. NO3 and NH4 uptake occur in parallel and can jointly affect diatom and other phytoplankton.

As the reviewer says, we fully acknowledge the important role that $NO_3^-$ declines have on diatom relative abundance at several places in the manuscript. However, we remain strong in our position that we have indeed isolated the effect of competition specifically for $NH_4^+$ on diatom relative abundance in our model experiments.

In the revised manuscript we have added a section dedicated to the N limitation formulation and described in more detail the modelling experiments, particularly model$_{compete}$. These additions should provide the detail required to convince the reviewer and readers of the paper that we have indeed isolated the effect of

competition for NH4 on the community composition of phytoplankton in our model experiments.

The new section called "Isolating the effect of competition for NH$_4$ reads:

Lines 187 – 237:
*"2.2.2 Isolating the effect of competition for $NH_4^+$*

[revised manuscript text omitted]

We sincerely hope that the review and other readers will be primed by this section to see that our modelling experiments have indeed isolated the effect of $NH_4^+$ on

phytoplankton community composition changes in our experiments.

**Specific comments**

Section 2.1: It is somewhat surprising that the model explicitly represents diatoms and nanoplankton, but not picoplankton, given the focus on competition between diatoms and cyanobacteria. The most abundant cyanobacteria in oligotrophic regions, Prochlorococcus and Synechococcus, fall within the picoplankton size range. But one could argue that the model's nanoplankton functionally encompasses both pico- and nano-plankton. A brief discussion of this issue and the potential limitations it introduces could be included.
We agree with the reviewer. We have included an additional sentence in this section:

Lines 114 – 116:
*"While the model does not strictly represent picophytoplankton, implicit variations in the average cell size of the nanophytoplankton type affect nutrient uptake dynamics and may therefore encompass some functionality of picophytoplankton in oligotrophic systems (Aumont et al., 2015)."*

The model description in the SI could be expanded for clarity. It would be helpful to include the equations for DIN uptake, as this is central to interpreting the results of decreased NO3 supply and understanding the distinction between model_control and model_compete experiments. Additionally, including the temperature dependence formulations for phytoplankton and zooplankton growth and grazing would help interpreting the warming-only experiments.
We completely agree and we have included an entirely new section on N limitation parameterisation (see answer above). Regarding temperature-dependent growth, we have also included another sentence in the methods section:

Lines 168 – 172:
*"In contrast, experiment "Warm" maintained the preindustrial climatological ocean state and atmospheric $CO_2$ at 284 ppm, but ensured that the ecosystem component saw increasing temperatures (T in °C) according to the RCP8.5 scenario, which scaled growth of phytoplankton types according to $1.066^T$ and heterotrophic activity (grazing and remineralisation) according to $1.079^T$ (Aumont et al., 2015)."*

The Tara analysis appears biased by the use of observations from depths shallower than 10 m only (lines 171-172). This likely skews the results towards phytoplankton communities adapted to relatively low NO3 and high NH4, in particular in oligotrophic regions, where phytoplankton are commonly found down to 100-200 m depths. This seems like a potentially important limitation, and could benefit from discussion. To my knowledge (but I may be mistaken), Tara Ocean also collected data from deep chlorophyll maxima. Why not including those data in the analysis, or at least consider them in a separate analysis?

We thank the reviewer for raising this important point. We agree that phytoplankton communities in oligotrophic regions often extend well below 10 m, and that samples from the deep chlorophyll maximum (DCM) may provide ecologically distinct information. However, our decision to focus on near-surface samples (<10 m) was guided by several practical and conceptual considerations.

First, the number of Tara Oceans samples from the DCM is substantially smaller than from the near-surface layer, which would considerably reduce statistical power in a global analysis—particularly when stratifying by environmental gradients or when performing region-specific comparisons.

Second, with respect to the $NH_4^+$:DIN ratio, DCM samples do not substantially expand the dynamic range of this ratio. In fact, $NH_4^+$:DIN ratios at the DCM tend to be consistently low, due to the combined effects of lower $NH_4^+$ concentrations and elevated $NO_3^-$ associated with the nitracline. In contrast, near-surface samples span a much broader and more variable range of $NH_4^+$:DIN values, especially across horizontal gradients in nutrient supply and ocean productivity. This variability is essential for detecting statistically robust relationships between nitrogen substrate ratios and shifts in phytoplankton community composition. By taking the more numerous near-surface samples, we encompass the full range of $NH_4^+$:DIN ratios.

We have added a statement to the revised manuscript clarifying this rationale and acknowledging the limitation explicitly.

Lines 387 – 389:

*"We exclusively used the data sets corresponding to surface samples (5-9 m depth) because of greater sampling coverage in the Tara Oceans dataset, which accesses a broad range of $NH_4^+$:DIN ratios spanning many ocean biomes/provinces."*

Section 2.5: the use of model-based fields in the analysis of observations makes me a bit uncomfortable, as it could introduce new, hard to control biases.
We completely share this sentiment because model fields are themselves highly uncertain, and as the reviewer states introduces new biases. Unfortunately, for the NH4:DIN ratio, NH4 is incredibly scarce as it is a very difficult measurement to make. We must therefore rely on the model to produce fields that can then be used to test our hypotheses.

While we cannot fully address this comment because there is not an observational product with full global coverage of NH4, we do provide a new section at the beginning of the results that is a model-data assessment of NH4 and NH4:DIN ratios in the ocean using the global dataset of NH4 that was compiled for this study.

Lines 337 – 365:

*"3.1 Assessment of modelled $NH_4^+$ and $NH_4^+$:DIN*

*Concentrations of 0.1 µM $NH_4^+$ or greater exist over continental shelves and in regions of strong mixing with high rates of primary production and subsequent heterotrophy. This accumulation of $NH_4^+$ in productive regions is reproduced by our model (Fig. 3a). In these eutrophic systems, high $NH_4^+$ co-occurs with high $NO_3^-$ concentrations, so $NH_4^+$ makes a small contribution to total DIN (Fig. 3b). These regions include the eastern tropical Pacific, eastern boundary upwelling systems, the northwest Indian Ocean, the subpolar gyres and the Southern Ocean (although we note that the model underestimates $NH_4^+$ concentrations in the Southern Ocean). In contrast, low $NH_4^+$ concentrations of less than 0.05 µM pervade the oligotrophic gyres of the lower latitudes. As these regions also display very low $NO_3^-$ concentrations, $NH_4^+$ makes up a much higher fraction of total DIN in both the observations and our model, with the $NH_4^+$ peak occurring deeper in the water column (Fig. S1).*

*Eutrophic upwelling systems and oligotrophic waters differed in the major sinks of $NH_4^+$ (Fig. 3c), consistent with available observations and constraints from theory. In eutrophic waters (here defined by surface nitrate > 1 µM), ammonia oxidation represented 49 ± 29 % (mean ± standard deviation) of $NH_4^+$ sinks, but this dropped to 32 ± 9 % in oligotrophic systems. Measured rates of ammonia oxidation showed a positive relationship with surface $NO_3^-$ concentrations and this was reproduced by the model (Fig. S2), indicating that ammonia oxidation was indeed a greater proportion of the overall $NH_4^+$ budget in eutrophic regions. In agreement, isotopic methods have shown that the bulk of nitrogen assimilated by phytoplankton in oligotrophic waters is recycled (Eppley and Peterson, 1979; Fawcett et al., 2011; Klawonn et al., 2019; Van Oostende et al., 2017; Wan et al., 2021), implying that most nitrogen cycling occurs without ammonia oxidation. Our model reproduces this feature of oligotrophic systems (Fig. 3c). Overall, the model shows good fidelity to the available observations of $NH_4^+$ concentrations, $NH_4^+$:DIN ratios, and rates of $NH_4^+$ cycling that we compiled for this study (Fig. 3; Fig. S1-S2).*

[Figure]

*Figure 3. Global patterns of $NH_4^+$ concentrations, its contribution to DIN in the euphotic zone, and $NH_4^+$ budgets. (a) The simulated maximum $NH_4^+$ concentration within the euphotic zone. The maximum was chosen to emphasise basin-scale variations. (b) Average values of the $NH_4^+$:DIN ratio. Modelled values are annual averages of the preindustrial control simulation between years 2081-2100. Observed values following linear interpolation between the surface and 200 metres depth are overlaid as coloured markers. Only those profiles with at least 3 data points within the upper 200 metres are shown. (c) Global mean ± standard deviations of $NH_4^+$ fluxes separated into eutrophic and oligotrophic regions. Sources of $NH_4^+$ are represented by positive values and sinks by negative values."*

This additional section helps build confidence in the model, at the very least in the broad spatial patterns seen across the open ocean.

Figure 2: I suggest including panel S1b, showing present-day NH4/DIN ratios with observations, in Fig. 2. This would reassure the reader that the model captures the basic N distribution patterns, and would help contextualize the changes shown in panel 2a (e.g., is a change by 10% large or small?) The text too could be more explicit , e.g., line 218, increase by 6%, could specify "from X% to Y%".
Please see answer above, as we have brought Fig. S1 into the main paper (now Fig.

3). We also added more explicit values of NH4:DIN changes.

Lines 368 – 371:
*"By the end of the 21st century (2081-2100), $NH_4^+$:DIN is projected to increase in over 98% of the upper ocean euphotic layer (Fig. 4a). On average (± standard deviation), the fraction of DIN present as $NH_4^+$ increased by 6 ± 6 % from a preindustrial average of 11.5 ± 11.0 % to 17.5 ± 14 %, with enrichment exceeding 20% in regions with pronounced DIN gradients, such as at the boundary between eutrophic and oligotrophic regimes."*

Lines 219-220: it would be interesting to report the % regenerated primary production and its change.
Great idea. We have included this in our revised manuscript.

Lines 371 – 374:
*"The enrichment of $NH_4^+$ caused an expansion of regenerated production across the ocean, such that $NH_4^+$ overtook $NO_3^-$ as the main nitrogen substrate for phytoplankton growth in an additional 10% (73% to 83%) of the ocean's area. Regenerated production also increased as a proportion of net primary production from 60% to 63%."*

The parameterizations for the pH dependence of ammonia oxidation are not well explained. What is the rationale behind including this pH dependence, and how were the specific functional forms shown in Fig. S5 and S7 chosen? Is it as simple as making the rate inversely proportional to the H+ concentration? (This seems the implication of the formulation invoking the pKa of NH4 dissociation.) What about the alternative formulation? The SI provides too little detail on these points, and more thorough explanation would help readers understanding the models sensitivities.
We agree with the reviewer that this was not communicated well in the previous version. In our revised manuscript, we have made efforts to communicate the pH effect on ammonia oxidation more explicitly, explaining that this relationship is data constrained.

Lines 172 – 175:
*"Experiment "OA" held the circulation and temperature effects on metabolism constant but involved the historical and future projected increase in atmospheric $CO_2$. This decreased pH and negatively affected rates of ammonia oxidation at a rate consistent with field measurements (Beman et al., 2011; Huesemann et al., 2002; Kitidis et al., 2011), specifically a loss of ~20% per 0.1 unit decrease in pH below 8.0 (Fig. S1)."*

Lines 387 – 389:
*"To accommodate some of this uncertainty, we performed an idealized experiment with a weaker relationship between pH and ammonia oxidation that still fit the measurements well but that enforced a 10% decline in ammonia oxidation per 0.1 pH decline rather than 20% (Fig. S6)."*

Line 238-240, "in eutrophic regions, where coincidentally, shifts from low to higher NH4:DIN would have the greatest ecological impact": This could be clarified. The phrasing is a bit confusing, as eutrophic regions are typically characterized by high NO3 concentrations. If NH4 becomes more dominant in these areas, does that imply they are no longer eutrophic? Or is the point that even in nutrient-rich waters, a shift in the form of available DIN from NO3 to NH4 could significantly affect community structure?

It is indeed the latter point you raise. We have clarified the sentence and logic. Thank you.

Lines 391 – 394:

*"Thus, whether pH declines have a strong or weak effect on ammonia oxidation did little to change $NH_4^+$:DIN ratios in eutrophic regions where $NO_3^-$ is abundant and where diatoms represent a larger proportion of the phytoplankton community, and where coincidentally, shifts from low to higher $NH_4^+$:DIN would have the greatest impact on community composition."*

Section 3.2, lines 258-270. The authors make the point that loss of diatoms was "driven by a combination of stimulated microbial metabolism (60%) and physical changes (40%), while ocean acidification had negligible effects". While this may be accurate at a global mean level, it risks giving the misleading impression that increased microbial metabolism is the dominant driver of diatom loss across the ocean, as compared to decreased NO3 supply. But is is not true almost anywhere in the ocean, where on a local basis  (Fig. S8) changes due to NO3 supply (or light availability at high latitudes) greatly exceed the low, but consistently negative effects of warming on metabolism. But because physical effects on nutrient supply and light availability are both positive and negative, they tend to cancel out when averaged globally. This distinction between local drivers and global mean effects is not clearly conveyed in the current discussion and should be more explicitly emphasized.

The reviewer is right to point out that the declines in diatoms on a regional sense are driven largely by declines in $NO_3^-$ caused by physical changes (Fig. S7). We have rewritten this paragraph to pay homage to this important effect:

Lines 467 – 483:

*"Our climate change simulations projected a future decline in the relative abundance of diatoms globally by an average of 3%, while local declines in the subantarctic, tropical, North Atlantic, North Pacific and Arctic Oceans sometimes exceeded 20% (Fig. 5a,b; Fig. S7). Our sensitivity experiments enabled an attribution of the major drivers, at least in a coarse-grained sense. At a global scale, the loss of diatom representation within marine communities in our model was driven by a combination of stimulated microbial metabolism (60% of full response in experiment "All") and physical changes (40% of full response in experiment "All"), while ocean acidification had negligible effects (Figure 5c; Fig. S7). Ocean acidification had negligible effects because it largely raised $NH_4^+$:DIN ratios in oligotrophic subtropical gyres where diatoms were already of low proportion (Fig. 4c; Fig. S5). Averaged across the low latitude ocean (40°S – 40°N), diatoms also declined by an*

*average of 3% driven by the same factors (60% microbial metabolism and 40% physical changes), while more dramatic but very regional declines of diatoms near or exceeding 20% were due primarily to physical changes (Fig. S7). These global and regional declines have been predicted previously and are widely accepted to be due to a decline in bulk nutrient availability in the upper ocean (Bopp et al., 2005), although the large effect of stimulated metabolism here suggests that top-down grazing pressure, which is accelerated by warming, may also play a role (Chen et al., 2012; Rohr et al., 2023). That said, stimulating metabolism also increases phytoplankton nutrient demand, which eventually leads to greater DIN limitation (Cherabier and Ferrière, 2022). We indeed appreciate that the reduction of diatoms from phytoplankton communities as simulated by models is due to nutrient losses, in particular declines in $NO_3^-$ (Kwiatkowski et al., 2020), and our simulations here, at least indirectly, are no different, since both nanophytoplankton and diatom biomass declined.''*

Lines 272-282. I find this paragraph unconvincing in its current framing. Since diatoms preferentially take up NO3 (given the DIN uptake formulation), it seems straightforward that a decline in NO3 supply and concentrations would reduce diatom production and abundance, without the need to shift the emphasis to increased competition for NH4, which would be a consequence of the change. The reduction in NO3 and the resulting increased reliance on (and potential competition for) NH4 seem more like two sides of the same coin, rather than distinct mechanisms. I'm not sure that NH4 competition is the most correct or useful framework for interesting the model changes. The authors end the paragraph by stating: "when NO3 concentrations decline, competition for NH4 increases, and declines in diatom relative abundance follow". But  is the middle step, "competition for NH4 increases", strictly necessary to explain the decline in diatoms? It might be more parsimonious to attribute the decline directly to reduced NO3 availability.

We hope that the new section (2.2.2 "Isolating the effect of competiton for $NH_4^+$") already shown in our answer above clarifies why our experiments isolate the effect of competition for $NH_4^+$. We do not argue against the reviewer that $NO_3^-$ is important, but we do state that the declines in $NO_3^-$ are important indirectly in that it preconditions intense competition for $NH_4^+$, which is the direct cause of 70% of the diatom declines in relative abundance. Our experiments with model$_{compete}$ isolate this effect of $NH_4^+$ competition.

The analogy (lines 277-278) should help understanding, here I found it confusing. If nutrients represent the volume in the bathtub, why is productivity described as the inflow? Productivity removes nutrients, it doesn't add them. And why is recycling represented as the outflow? Conceptually, recycling returns nutrients to the system. Also, at steady state, one would expect inflow and outflow to be in balance. This needs some rethinking or clarification.

We agree that the previous sentence was confusing, and we have rewritten this sentence.

Lines 490 – 491:
*"This is akin to the bathtub analogy, where different volumes (i.e, nutrient concentrations)*

*can result by varying the inflow (i.e., recycling) even when the outflow is constant (productivity)."*

Lines 291-294, "It is therefore possible that reductions in NO3 and resulting competition for NO3 was a major contributor to the losses of diatoms from the phytoplankton community in our simulations". This seems like a straightforward explanation for diatom decline, given diatom's functional dependence on DIN forms and the NO3 declines. But its placement here appears to undercut the argument made a few paragraphs earlier that emphasized competition fro NH4.
This sentence has been removed.

Lines 294-304, the model_compete experiment is interesting, but I see it more as highlighting another side of the same coin, rather than challenging the idea that declining NO3 is the primary driver of diatom decline. Of course if diatoms had a stronger affinity for NH4, they would fare better under reduced NO3. But this doesn't change the underlying cause of their decline, which still originates from the reduction in NO3.
We agree completely with the reviewer. The loss in $NO_3^-$ preconditions the community to compete for $NH_4^+$. Thus $NO_3^-$ losses are a cause of diatom losses, but are in part an indirect cause, since our experiments show that if diatoms were competitive for $NH_4^+$ they would fare much better. We hope that the new detail, in particular section 2.2.2, has clarified this point.

Also, please see our responses below with regard to this issue.

Line 311, "Physical changes no longer exerted a global negative effect on their total nor relative abundance". The global mean masks large regional variability, where large positive and negative changes partially compensate each other. As shown in Fig. S8, physical changes are the main local driver, especially in the Southern Ocean, and in fact, across much of the ocean on a point-by-point basis. This nuance should be acknowledged more clearly.
We agree with the reviewer and have added a clause to this sentence to add nuance to the sentence.

Lines 561 – 563:
*"Physical changes, while important regionally, no longer exerted a global negative effect on their total nor relative abundance (blue line in Fig. 5f), while the negative effect of elevated microbial metabolism on relative abundance was ameliorated by 25% (Fig. 5f; Fig. S7-S8)."*

Lines 238-240, "determined that a large fraction of the projected declines in diatom relative abundance are due to their competitive exclusion by other phytoplankton in regions where NH4 becomes more important as a nitrogen source": I am unconvinced that the authors have "determined" that the diatom decline is caused by competitive exclusion, and not by the overall decline in NO3.
We can attribute this decline in the modelled diatom relative abundance to

competition for $NH_4^+$ and their competitive disadvantage in this respect, at least in the model, due to our sensitivity experiments with $model_{compete}$.

Section 3.3.1 and Fig. 4: this section provides a stimulating and valuable set of analyses and diagnostics that could serve as a useful benchmark for models. More ocean biogeochemical models should adopt this type of diagnostic approach as a standard practice for evaluating nutrient dynamics and phytoplankton competition. We are pleased that the reviewer also values this analysis highly.

Lines 369-371: does it matter than in PISCES the half saturation constants for DIN uptake are not constant but a function of the phytoplankton biomass P? What are "typical conditions"? To help the reader, the DIN uptake functional forms used by the model should be presented in the SI.
The reviewer is right that the half-saturation coefficients for diatoms and nanophytoplankton are not static but change as a function of phytoplankton biomass as a way to capture changes in the community mean cell size. However, since diatoms half-saturation coefficients are prescribed as being 3-time greater than nano-phytoplankton, they are always greater, meaning that the unique N limitation parameterisation now shown in Fig. 2 will always be the case. We have presented the N limitation functional forms now in section 2.2.2.

Section 3.3.2: this is a stimulating analysis—though at times it felt substantial enough to warrant its own standalone paper. Figures S10-S16 are rich with interesting information, but also dense. I wonder if "goodness of fit" metrics (some o which may be reported in Table S1?) could be included directly on the figures, so that the reader can quickly evaluate the skill of different models.
We agree with the reviewer and thank them for their positive and constructive comments about this analysis, which we also see as highly valuable. All goodness of fit information is indeed presented in Table S1, which the reader will be able to reference. An easy way to understand if a relationship is significant is whether the confidence intervals do not include zero. For example, in Fig. S9, Silicic acid has no significant predictive power for diatom relative abundance for both the 18S rRNA and *psbO* gene counts, but $NH_4^+$:DIN does. We refer the interested reader to Table S1 for the actual degree of significance and the amount of deviance explained by the predictor in question.

Fig. 5: I got a bit lost in the interpretation of the figure. Not being familiar with GAMs, I am confused by the y-axis, "GAM residuals", which could be clarified. Also, related to the threshold of 4%; in Fig. 5 nothing specifically makes the 4% threshold stand out. The text points to "rapid losses of diatoms as NH4:DIN became greater than 4%", but I struggle to see anything special in this threshold, and overall it seems an arbitrary choice.

We added two new sentences to the legend in Figure 7 explaining what the GAM residuals mean and also why a 4% threshold was chosen.

Lines 819 – 820:

*"When GAM residuals are positive this suggests that diatoms do better than predicted by a GAM without the $NH_4^+$:DIN ratio as a predictor, and vice versa."*

Lines 823 – 824:
*"The vertical dotted line delineates when $NH_4^+$ is 4% of DIN, which aligns with the point at which community primary production switches from predominantly $NO_3^-$-fuelled to $NH_4^+$-fuelled (Fig. 6)."*

We also point the reviewer to this sentence *"This threshold, where $NH_4^+$ becomes 4% of total nitrogen stocks, aligned with the point at which primary production becomes dominated by regenerated production (Fig. 6)."* Starting on line 841.

Lines 449-453: points (1) and (3) are general knowledge; point (2) could be as well phrased as "how diatoms are outcompeted as less primary production is fueled by external NO3 inputs", which would actually be closer to the mechanistic changes at play.
We agree and have changed this sentence in line with the reviewer's suggestion.

Section 3.3.4, "The confounding effect of NO3": framing the role of NO3 as a confounding factor underplays its central role in controlling diatom growth, and seems an example of reverse causation and post hoc reasoning. Given the well-established importance of physical changes on NO3 supply across the ocean, reductions in NO3 should be taken as the default or "null" mechanisms to explain the diatom declines, rather than a confounding effect. The fact that the whole statistical analysis could be based on NO3 instead of NH4:DIN as a covariate seem to undermine the entire argument made here.
We agree with the reviewer that $NO_3^-$ is indeed important in that its loss from the environment forces the phytoplankton species to compete for $NH_4^+$. We also agree that the loss of $NO_3^-$ and increased competition for $NH_4^+$ are "two sides of the same coin", so to speak. And we do hope that this is made clear in our revised manuscript.

We do, however, remain loyal to what the model$_{compete}$ results tell us, which is that competition for $NH_4^+$ is the more direct cause of much of the diatom declines. This detail should be clearer in our revised manuscript due to the addition of section 2.2.2.

In this section and the revised results sections we show that $NO_3^-$ concentrations can (and do!) decline in the experiments conducted with both model$_{control}$ and model$_{compete}$, but the only difference is that competition for $NH_4^+$ disadvantages diatoms (model$_{control}$) or it doesn't (model$_{compete}$). The model output from our

experiments is therefore clear in this regard. We note here for clarity that we do not discount at any stage the importance of other processes, nutrients, etc., which is evident in our discussion of the changes in diatom relative abundance in the Southern Ocean (Lines 570-575), and we fully acknowledge the importance of these other drivers in our discussion of the results.

We have renamed this section "The indirect effect of $NO_3^-$" and we have edited it in the revised manuscript to more clearly communicate the points above.

Lines 862 – 881:
*"We fully acknowledge that $NH_4^+$:DIN ratios covary strongly with $NO_3^-$ concentrations. Most of the projected increases in $NH_4^+$:DIN we report here are due to circulation changes that limit $NO_3^-$ injection from subsurface waters into surface waters (Fig. S5). Also, our GAM analysis of the Tara Oceans data could easily be replicated by replacing the $NH_4^+$:DIN ratio with $NO_3^-$ concentration as a key predictor. Indeed, this analysis showed similar results, with $NO_3^-$ being an equally strong predictor of diatom relative abundance as $NH_4^+$:DIN. We therefore cannot discount a direct effect of $NO_3^-$ on diatom relative abundance in the Tara Oceans observations.*

*In our biogeochemical model, however, we can diagnose whether diatom relative abundance changes are directly due to competition for $NO_3^-$ or $NH_4^+$. This allows us to assess whether $NO_3^-$ concentration or the $NH_4^+$:DIN ratio are more appropriate as a predictor of diatom relative abundance. The importance of $NH_4^+$ is exemplified by the fact that the negative relationship between $NH_4^+$:DIN and diatom relative abundance was reversed in model$_{compete}$ (black dotted line in Fig. 7a). Now positive rather than negative, this relationship differs statistically from that predicted from Tara Oceans data (Figure 7b-e).*

*This suggests that competition for $NH_4^+$ directly controls diatom relative abundance in our model. We fully acknowledge that a scarcity of $NO_3^-$ is a major cause of $NH_4^+$ enrichment in our experiments because it drives competition for $NH_4^+$. However, we wish to emphasize that a potentially important mechanism of diatom decline in the community is due to their poor competitive ability for growth on $NH_4^+$, not directly because of decreases in $NO_3^-$. Decreases in $NO_3^-$ certainly affect diatom growth, but, in our model, they mostly do so indirectly by shifting the regime towards intense competition for $NH_4^+$. Given the statistical similarity between the in situ (Tara Oceans) and in silico (model$_{control}$) relationships (Fig. 7) and the dissimilarity in model$_{compete}$, this points to $NH_4^+$:DIN as a key underlying driver of diatom relative abundance in the world ocean."*

Line 473, "We therefore suggest that competition for NH4 directly controls diatom relative abundance": this feels like a forced interpretation and a leap that is not fully supported by the evidence presented.
Please see above response.

Line 477-478, "Decreases in NO3 certainly affect diatom growth, but we propose that they mostly do so indirectly by shifting the regime towards intense competition for NH4." Another sentence that seems to overstate the authors' interpretation

without enough supporting evidence, given the direct role of NO3 in diatom growth, which is explicitly built into the model's functional formulation.
Please see above response.

Line 486-487, "but we recast the attribution of change in terms of competitive exclusion for NH4, rather than bulk nutrient declines": see above.
Please see above response.

**Technical comments:**

Line 66, "become a self-sustaining regime": I would rephrase, partly because the message is a bit unclear, partly because it is hard to imagine a case where primary production in the euphotic zone does not involve an external supply of NO3, even in the most oligotrophic regions (were export still occurs and must be ultimately balanced by external nutrient supply).
Line 68 :
*"Due to the intense competition for $NH_4^+$ and resulting shifts towards smaller phytoplankton taxa that are more rapidly recycled in the upper water column, the relative enrichment in $NH_4^+$ may become a self-sustaining regime unless new inputs of $NO_3^-$ are sufficient to reverse it."*

Line 45, "which would work to use up …" this sentence is a bit obscure, please clarify.
Line 46 – 49:
*"One theory posits that their ecological success in turbulent, high $NO_3^-$ environments (Margalef, 1978) may be due to a capacity to store $NO_3^-$ in their vacuoles and then rapidly reduce it when they experience sudden increases in light, which would position diatoms to rapidly consume any excess reductant that would otherwise retard growth (Glibert et al., 2016a; Lomas and Glibert, 1999; Parker and Armbrust, 2005)."*

Line 65, "there are numerous localized studies that showcase how phytoplankton taxa shift in response to changes in the composition of DIN": it would be useful to add some references for these studies.
Added.

Line 74: remove repeated "enrichment".
Removed.
Line 129, "defined as those depths where total phytoplankton biomass was greater than 0.1 mmol C m-3." This is a bit of an unorthodox definition of euphotic zone, perhaps clarify the rationale and reassure it is generally in line with other common light-based definitions (typically, 1% light levels)
We have changed this from euphotic zone to "upper ocean where primary production is active".

Lines 181 – 183:
*"We calculated changes at each grid cell by averaging over the upper ocean where primary*

*production was active, which was defined as those depths where total phytoplankton biomass was greater than 0.1 mmol C m$^{-3}$.”*

Lines 146-149: here a relevant reference could be Tang, et al., 2023, Earth System Science Data, which presents a compilation of nitrification rates. (Presumably data used there and compiled in Tang et al. overlap.)
We agree. We, however, didn't use the Tang dataset because our study predates this dataset.

Line 151, I appreciate the authors reporting the units for primary production; for consistency, they could be added for the other major variables discussed (e.g., ammonia oxidation rates).
Rectified.

Line 194: maybe provide some more detail on the spline approach. What is k, in k=3? What is "spline complexity"? Etc.
We have provided a greater description regarding these choices:

Lines 296 – 326:
*"Where Y is the predicted response, $\alpha$ is the intercept, $s_n(x_n)$ represents a smooth function (specifically the n$^{th}$ thin-plate spline) fitted to the n$^{th}$ predictor variable $x_n$, and $\varepsilon$ is the model error. Thin-plate splines are flexible and widely used as a smoothing method within GAMs that allow for non-linear relationships between predictors and response variables and do not require specificity around a functional form. They are well suited to handling ecological data where relationships are often non-linear and non-parametric. Predictor variables were mixed-layer depth (m), phosphate (µM), silicate (µM), dissolved iron (µM), and the $NH_4^+$:DIN ratio. Mixed layer depth, phosphate and silicate was measured in situ at the sample locations of Tara Oceans, while dissolved iron and $NH_4^+$:DIN ratios were provided by the model at the same location and month of sampling, since measurements of these properties are scarce. In addition, phosphate and silicate concentrations were available as interpolated products from the World Ocean Atlas (Garcia et al., 2019). An alternative estimate of $NH_4^+$:DIN ratios was provided by the Darwin model (Follows et al., 2007). Predictor variables from models and World Ocean Atlas were extracted at the locations and months of sampling and different combinations of in situ and modelled variables were used to build GAMs. Mixed-layer depth, nutrients (phosphate, silicate and $NH_4^+$:DIN) and the relative abundance of phytoplankton taxa were log$_{10}$-transformed prior to model building to ensure homogeneity of variance.*

*Before model testing, we calculated the variance inflation factors (VIFs) of independent variables to avoid multi-collinearity. All covariate VIFs were < 3, which indicates minimal multicollinearity. GAMs were computed using a low spline complexity (k = 3) that prevented overfitting and constrained the smooth functions represent only broad-scale trends in the data."*

Line 289: "thid" —> "this".
Corrected.

Line 307: should the reference be to Fig. 3e?
Yes. Now it is referenced to Fig. 5d-f.

Line 322, "by the in the", remove "in the".
Corrected.

Line 356: since the exponent is fractional, I would call the function a "fractional-order" Monod function, not a quadratic (where the implied exponent is 2).
Corrected.

Lines 358-359: add the units for the half saturation constants.
Added.

Line 493: "albiet" —> "albeit".
Corrected

- Line 496: "ellaboratiung" —> "elaborating".
 Corrected

Fig. S1 could also show total DIN, so that one has a complete view of the controls on the NH4/DIN ratio distribution.
This is now Fig. 3 in the main text. We thank the reviewer for the suggestion, but maintain our presentation of NH4 and the NH4:DIN ratio, since with these two pieces of information it is possible to understand DIN.

Fig. S8, right column: I'm confused by the units, the values go up to 0.2, but the units say %. Shouldn't Fig. S8c be the same as S3a, were value go up to 20%? The figures also look a bit different, but perhaps it's the contouring that make them look different.
Corrected the units. The reviewer is right and this was a typo.